# Impact Factors of Industrial Pollution and Carbon Reduction under the "Dual Carbon" Target: A Case Study of Urban Aggregation in the Pearl River Delta and Yangtze River Delta

Xiaoyi Wen [1], Shangjiu Wang [1,*] , Shaoyong Li [1], Liang Cheng [2], Keqiang Li [3,*], Qing Zheng [1] and Baoreng Zhang [1]

1   School of Mathematics and Statistics, Shaoguan University, Shaoguan 512005, China; wlapr10@163.com (X.W.); lishaoyongok@sohu.com (S.L.)
2   School of Political Science and Law, Shaoguan University, Shaoguan 512005, China
3   College of Digital Technology and Engineering, Ningbo University of Finance and Economics, Ningbo 315175, China
*   Correspondence: wangshangjiu@163.com (S.W.); likeqiang000@sina.com (K.L.)

**Abstract:** China is facing pressure to reduce carbon emissions and control pollution. Promoting the synergy between pollution reduction and carbon reduction has become an inevitable choice to achieve the construction of a beautiful China and meet the dual carbon target. This study examines the main factors influencing industrial pollution and carbon reduction in the Pearl River Delta (PRD) and Yangtze River Delta (YRD) urban agglomerations based on data on industrial $CO_2$ and local air pollutants (LAP) from 2002 to 2021, using the random forest regression model. The results indicate that (1) industrial $CO_2$ emissions have increased overall, while intensity has decreased. Additionally, both industrial LAP emissions and intensity have decreased. (2) The main factor influencing industrial $CO_2$ and LAP emissions is the proportion of industrial value added above the scale. Additionally, the proportion of R&D internal expenditure in GDP and total trade imports and exports are the main influencing factors of industrial $CO_2$ emissions. The industrial fume and dust removal rate mainly affects industrial LAP emissions. (3) There is a clear non-linear relationship between industrial $CO_2$- and LAP-influencing factors and emissions, which can be attributed to the scale effect factor, the lagging effect of R&D expenditure, and the inappropriate treatment of the "three wastes" by relevant departments that is non-linear. The urban agglomerations of PRD and YRD should prioritize the reduction of carbon emissions, upgrading and transforming their industrial structures, promoting the impact of foreign trade on pollution and carbon reduction, and achieving a balance between sustainable economic development and environmental protection.

**Keywords:** pollution and carbon reduction; random forest regression model; Pearl River Delta urban agglomeration; Yangtze River Delta urban agglomeration

## 1. Introduction

Energy and environmental issues are crucial factors in global economic and social development. There is an international consensus to address climate change and develop a low-carbon economy proactively. China, being the largest country in energy consumption and carbon emissions, has a significant responsibility to reduce pollution and carbon emissions. On 22 September 2020, Xi Jinping, General Secretary of the Communist Party of China Central Committee, Chinese President and Chairman of the Central Military Commission, delivered a speech at the general debate of the 75th United Nations General Assembly. He announced that China aims to peak carbon dioxide emissions by 2030 and achieve carbon neutrality by 2060. This marks China's inaugural commitment to the United Nations stage, a significant development in global climate governance. The "dual carbon" goal was subsequently incorporated into China's 14th Five-Year Plan and

2035 Vision Outline. On 26 February 2021, the National Development and Reform Commission (NDRC), the State-owned Assets Supervision and Administration Commission of the State Council (SASAC), and others issued the "Guiding Opinions on Strengthening Green and Low-Carbon Transformation and Development". The goal proposed is to achieve peak carbon dioxide emissions before 2030. The "dual carbon" goal aims to promote China's high-quality development and the construction of an ecological civilization. This concept emphasizes the harmonious coexistence of human beings with nature, sustainable development, and environmental protection. It is a comprehensive development concept in China. Additionally, countries around the world are implementing various 'dual carbon' policies to address climate change and achieve carbon peak and neutrality. For instance, in the United States, these policies involve rejoining the Paris Agreement, setting ambitious carbon-neutral targets for 2050, and making significant investments in clean energy and infrastructure. The administration's policy emphasizes comprehensive climate action, including concerns about specific sector objectives, policy reversals, and cooperation with domestic and international partners [1]. The European Union's "Green Deal" is an ambitious plan to transform its member states into carbon-neutral economies by 2050. This plan addresses the energy transition, promotion of renewable energy, and industrial innovation [2]. Despite its energy-intensive industries, India is aggressively pursuing carbon reduction targets and investing in renewable energy and clean technologies for sustainable development [3]. These policies reflect the global consensus on cooperation in tackling climate change and highlight the challenges faced by countries in achieving carbon neutrality.

Scholars have researched the factors influencing pollution and carbon reduction from various perspectives to achieve the goal of reducing pollution and carbon emissions. Research has been conducted at the national level [4–7], the local level [8,9], and the industry sector level [10,11]. The research method perspective is mainly attributed to the decomposition analysis method, which includes the index decomposition analysis method (IDA) and the structural decomposition analysis method (SDA) [12,13]. For instance, Yan et al. (2019) used the IDA method to identify the main influencing factors of the overall change in $CO_2$ emissions from provincial thermal power generation in China [14]. Wang et al. (2019) employed the SDA method to investigate the driving factors behind changes in carbon emissions resulting from domestic trade in China at a regional scale [15]. Other scholars have also utilized econometric models to examine the factors that influence pollution and carbon reduction [16]. For instance, Xian et al. (2019) evaluated the synergistic effect and mechanism of the carbon trading pilot policy on carbon and air pollutant emissions in China using the double difference model (DID). This study covered the power, industry, transportation, and residential sectors [17]. Zhu et al. (2020) analyzed the factors influencing hidden carbon emissions in China's construction industry using the extended STIRPAT mode [18]. Lin et al. (year not provided) used the Spatial Durbin Error Model (SDEM) to investigate the socioeconomic factors that can jointly impact $CO_2$ and PM2.5 emissions [19]. Scholars have identified the following aspects as the main influencing factors of pollution and carbon reduction: scale effect, structural effect, technical effect, governance factors, trade factors, and population factors [20–22].

In general, the following shortcomings persist: (1) The reduction of pollution and carbon emissions at the industrial level is a crucial aspect of the "dual carbon" program. However, the existing research has largely neglected this area, with most studies focusing on the factors that influence the reduction of greenhouse gases or air pollutants. This is insufficient to meet current development needs. (2) To date, the research on the factors that influence the reduction of pollution and carbon in the urban agglomerations of the Pearl River Delta and Yangtze River Delta has been inadequate. A further exploration of the YRD urban agglomeration is necessary, along with additional research on inter-provincial heterogeneity. This will provide local governments with the necessary information to formulate differentiated pollution and carbon reduction policies.

Therefore, we analyze data on industrial $CO_2$ and LAP emissions from 2002 to 2021 and use a random forest regression model to systematically study the factors influencing pollution and carbon reduction in the PRD and YRD urban agglomerations. The main objectives are as follows: (1) The aim of this study is to identify the main factors that influence industrial pollution reduction and carbon reduction in the PRD and YRD urban agglomerations. (2) Additionally, we aim to explore the temporal dynamic evolution patterns of industrial $CO_2$ and LAP emissions and intensities in these areas. (3) The results of this study will provide a more comprehensive and precise reference for the formulation of environmental protection policies for the PRD and YRD urban agglomerations. This approach contributes to a comprehensive understanding of the interactions between various factors in pollution and carbon reduction. It provides a scientific basis for future sustainable urban development.

This paper presents three new contributions. Firstly, it discusses pollution reduction and carbon reduction in the same framework. Secondly, it highlights the significance of fully utilizing the synergistic effects of the PRD and YRD urban agglomerations in industrial pollution reduction and carbon reduction. Finally, it emphasizes that the PRD and YRD urban agglomerations play a key role in China's economic and social development while simultaneously facing unique environmental challenges. This choice considers several factors, including the concentration of economic activities, differences in energy mixes, geographical characteristics, and potential policy implementations. The two regions share similarities in terms of high economic development and urbanization, but differ in their geographic location and industrial composition. Additionally, this thesis aims to provide information on the similarities and differences between the selected clusters in terms of industrial $CO_2$ and LAP emission reductions, while acknowledging the potential inter-provincial heterogeneity of the two clusters, including differences in environmental policies, resource distribution, and economic status. Thirdly, it provides a methodology to investigate how to achieve the synergistic development of pollution reduction and carbon reduction in multiple aspects, including energy, industrial structure, production technology, science, technology and innovation, and trade.

## 2. Data Sources and Research Methods
### 2.1. Data Sources and Variable Descriptions

Data on the indicators for the regions from 2002 to 2021 were collected from various sources, including the China Energy Statistical Yearbook, China Environ mental Statistical Yearbook, China Urban Greenhouse Gas Studio Platform, China Economic and Social Big Data Research Platform, and the statistical yearbooks of each province and city in the PRD and YRD urban agglomerations. The dataset organizes the data for each indicator, as well as the industrial $CO_2$ and LAP emissions of each city in the PRD (9) and YRD urban agglomerations (27), making it convenient for later model building and data reading.

Through basic statistical analysis, outliers and missing values are identified and addressed using linear interpolation and the linear trend of neighboring points.

### 2.1.1. Measurement of Industrial Carbon Dioxide ($CO_2$) Emissions

This paper examines five fossil energy sources: gasoline, diesel oil, raw coal, coke, and natural gas. The $CO_2$ emissions resulting from the combustion of these sources are measured using the emission factor method. Based on these measurements, the carbon emissions at the industrial level are calculated according to the $CO_2$ emission factors. One of the most commonly used methods for carbon emission accounting in academic research is to convert selected fossil energy sources into standard coal and measure them using a specific carbon emission factor. However, there are variations in the criteria used by different organizations and scholars in the determination of the carbon emission factor. This

thesis uses the coefficients from the Guidelines for the Preparation of Provincial Greenhouse Gas Inventories (Trial) to measure carbon emissions. The formula is as follows [23]:

$$CO_2 = A_i \times E_i, \tag{1}$$

where $A_i$ represents the fuel consumption of the fossil energy source of the ith species. $E_i$ represents the emission factor of the ith fossil energy source, as shown in Table 1.

**Table 1.** Carbon dioxide ($CO_2$) emission factors and relevant parameters for fuels.

| Energy Varieties | Average Net Calorific Value [(1)] | Carbon Oxidation Rate [(2)] | Carbon Emission Factor [(3)] | $CO_2$ Emission Factor [(4)] |
|---|---|---|---|---|
| Gasoline | 43,070 kJ/m$^3$ | 0.98 | 19.1 tc/TJ | 2.9251 kJ/m$^3$ |
| Diesel oil | 42,652 kJ/m$^3$ | 0.98 | 20.2 tc/TJ | 3.0959 kJ/m$^3$ |
| Raw coal | 20,908 kJ/m$^3$ | 0.94 | 25.8 tc/TJ | 1.9003 kJ/m$^3$ |
| Coke | 28,435 kJ/m$^3$ | 0.93 | 29.2 tc/TJ | 2.8604 kJ/m$^3$ |
| Natural gas | 38,931 kJ/m$^3$ | 0.99 | 15.3 tc/TJ | 2.1621 kJ/m$^3$ |

Note: (1) Average net calorific value (NCV) data from the IPCC Guidelines (2006). (2) Carbon oxidation rate is denoted by $\lambda$. (3) Carbon emission coefficient ($\theta$) data from "Guidelines for Provincial Greenhouse Gas Inventory Preparation (Trial)" (2010). (4) $CO_2$ emission factor: $CEF = NCV \times \lambda \times \theta \times 10^{-6} \times \left( \frac{44}{12} \right)$ in kJ/m$^3$.

### 2.1.2. Measurement of Emissions of Industrial Local Atmospheric Pollutants (LAP)

Based on the analyses, industrial sulfur dioxide ($SO_2$), nitrogen oxide ($NO_X$), and soot (powder) dust (PM) were selected to measure the industrial LAP emissions in this thesis. The collected data were processed, and the industrial LAP emissions were calculated with the following formula [24]:

$$LAP = \alpha Q_{SO_2} + \beta Q_{NO_x} + \gamma Q_{PM} \tag{2}$$

where $Q_{SO_2}$, $Q_{NO_x}$, and $Q_{PM}$ represent the emissions of sulfur dioxide, nitrogen oxide, and smoke (dust) in a region, respectively. The air pollutant volume coefficients specified in the Environmental Protection Tax Law (2018) are used to convert $SO_2$, $NO_x$, and PM to industrial LAP emissions, represented by $\alpha$, $\beta$, and $\gamma$, respectively.

### 2.1.3. Indicator System

To select the main influencing factors of industrial $CO_2$ and LAP reduction in PRD and YRD urban agglomerations, it is necessary to consider several aspects. Industrial production activities are a significant source of industrial $CO_2$ and LAP emissions; therefore, their scale is closely related to carbon reduction [25]. China's increasing coal-based energy consumption is a crucial factor in the rise of industrial $CO_2$ and LAP emissions. China's industrial sector dominates the country's energy consumption, resulting in high energy usage, pollution, and carbon emissions. Therefore, optimizing the industrial structure can significantly contribute to reducing pollution and carbon emissions [26]. Technological progress, environmental protection, and management of the increase in investments will have a greater positive effect on reducing pollution and carbon reduction [27]. The impact of import and export on China's carbon emissions is asymmetric, with import trade favoring the reduction of carbon emissions and export trade being the opposite [28]. Importing high-quality final and intermediate products can enhance the technological content of our production activities and reduce our country's carbon emissions [29]. Based on previous research results, combined with the actual situation of PRD and YRD urban agglomerations, this thesis selects a total of eight indicators in five dimensions, namely scale effect factor, structural effect factor, technology effect factor, governance effect factor, and trade effect factor, as the explanatory variables of the model. Based on previous research results, combined with the actual situation of PRD and YRD urban agglomerations, this thesis selects a total of eight indicators in five dimensions, namely scale effect factor, structural

effect factor, technology effect factor, governance effect factor, and trade effect factor, as the explanatory variables of the model, as shown in Table 2:

**Table 2.** Analysis index system of industrial pollution and carbon reduction impact factors.

| Indicator Dimension | Specific Indicators | Number | Statistical Methods | Unit |
|---|---|---|---|---|
| Scale effect factors | Industrial value added above the scale | SC1 | Calculation of comparable price using 2000 as the base period | CNY 100 million |
| Structural effect factors | The proportion of industrial value added in GDP | ST1 | Industrial value added/GDP | % |
| Technology effect factors | The proportion of R&D internal expenditure in GDP | TE1 | R&D internal expenditure/GDP | % |
| | Energy consumption per CNY 10,000 of industrial value added | TE2 | Industrial energy consumption/industrial value added | tce/CNY 10,000 |
| Governance effect factors | The proportion of environmental investment in GDP | GO1 | Investment in energy conservation and environmental protection/GDP | % |
| | Industrial sulfur dioxide removal rate | GO2 | Industrial sulfur dioxide removal/generation | % |
| | Industrial fume and dust removal rate | GO3 | Industrial fume and dust removal/generation | % |
| Trade effect factors | Total trade imports and exports | TR1 | Total trade imports + total trade exports | CNY 100 million |

(1) Scale effect factors

The scale effect, also known as scale economy, refers to the impact of economic progress on industrial $CO_2$ and LAP emissions. This influence process is mainly divided into two stages. Within a certain range, expanding the industrial scale can reduce production costs and improve production efficiency, thereby increasing energy consumption and emissions. However, it is important to note that beyond a certain range; the continuous expansion of industrial scale may hinder production activities and result in decreased emissions [30].

(2) Structural effect factors

The structural effects typically refer to the impact of changes in energy and industrial structure on industrial $CO_2$ and LAP emissions. Emissions are likely to decrease when industrial departments implement measures to optimize energy structure, such as industrial structure optimization or reducing the use of fossil fuels in favor of new energy sources [31].

(3) Technology effect factors

The technical effect refers to the impact of advancements in production technology within the industrial sector on $CO_2$ and LAP emissions. This includes both the scale and intensity effects of such progress [32]. The opposite is also true. In general, progress in industrial technology leads to increased energy utilization and decreased energy consumption, which in turn reduces emissions.

(4) Governance effect factors

The governance effect refers to the effect of the increase in national financial expenditure on environmental governance on industrial $CO_2$ and LAP emissions. This effect can be divided into two aspects: Financial support for energy-saving and environmental protection projects can encourage more industrial sectors to promote energy saving and pollution reduction. Additionally, qualified enterprises can enjoy tax incentives related to environmental protection according to relevant policies, which will also encourage enterprises to upgrade sewage treatment equipment. It is important to note that the expenditure on environmental governance is mostly granted to enterprises in the form of policy subsidies. However, it is crucial to maintain a balanced approach to avoid bias towards any particular sector. This allows qualified enterprises to benefit from tax incentives related to environmental protection, which in turn encourages them to upgrade their sewage treatment equipment. These measures can effectively reduce emissions [33].

(5) Trade effect factors

The trade effect refers to the effect of increased trade flows between home and foreign countries on industrial $CO_2$ and LAP emissions. In general, an increase in trade exports is associated with an increase in the intensity of production in the industrial sector and a further increase in emissions, and vice versa. At the same time, trade exports can further offset structural and technological effects [20].

### 2.2. Study Area

This paper takes the PRD and YRD urban agglomerations as research areas. Located in the south and east of China, these two urban agglomerations are rich in resources and superior geographical advantages, which make them an important engine of China's economic development. The specific cities are shown in Table 3.

**Table 3.** The study area.

| Region | Number of Cities | Specific Cities |
|---|---|---|
| The PRD urban agglomerations | 9 | Guangzhou, Foshan, Zhaoqing, Shenzhen, Dongguan, Huizhou, Zhuhai, Zhongshan, Jiangmen |
| The YRD urban agglomerations | 27 | Shanghai, Nanjing, Wuxi, Changzhou, Suzhou, Nantong, Yancheng, Yangzhou, Zhenjiang, Taizhou, Hangzhou, Wenzhou, Ningbo, Jiaxing, Huzhou, Shaoxing, Jinhua, Zhoushan, Taizhou, Hefei, Wuhu, Maanshan, Tongling, Anqing, Chuzhou, Chizhou, Xuancheng; |

### 2.3. Model Construction

The random forest regression model is a fusion algorithm based on a decision tree classification proposed by Breiman in 2001 [34]. The random forest model has significant advantages:

(1) It has strong processing capability for high-dimensional and large-scale data.

(2) Strong system promotion ability, which can effectively reduce the risk of system overfitting.

(3) Ability to handle missing values and outliers.

(4) Better fitting effect for non-linear correlation data.

Random forest regression [34,35] is a model combining multiple decision trees $\{g(x, \theta_t), t \in [1, T]\}$, where x is the dependent variable and $\theta_t$ is an independent and identically distributed random vector. By integrally learning each decision tree $\{g(x, \theta_t)\}$, it is averaged as a regression prediction. The calculation is shown in the following equation:

$$\text{Regression} = \Sigma_{t=i}^{T} \{g(x, \theta_t)\}, \tag{3}$$

where T represents the number of decision trees generated.

The specific flow of the random forest regression modeling approach is shown in Figure 1.

This thesis utilized the scikit-learn machine learning library, which is integrated with Python, to develop a random forest regression model. The model aimed to predict industrial $CO_2$ and LAP emissions of PRD and YRD urban agglomerations using five dimensions of influencing factors, consisting of a total of eight indicators, as input variables.

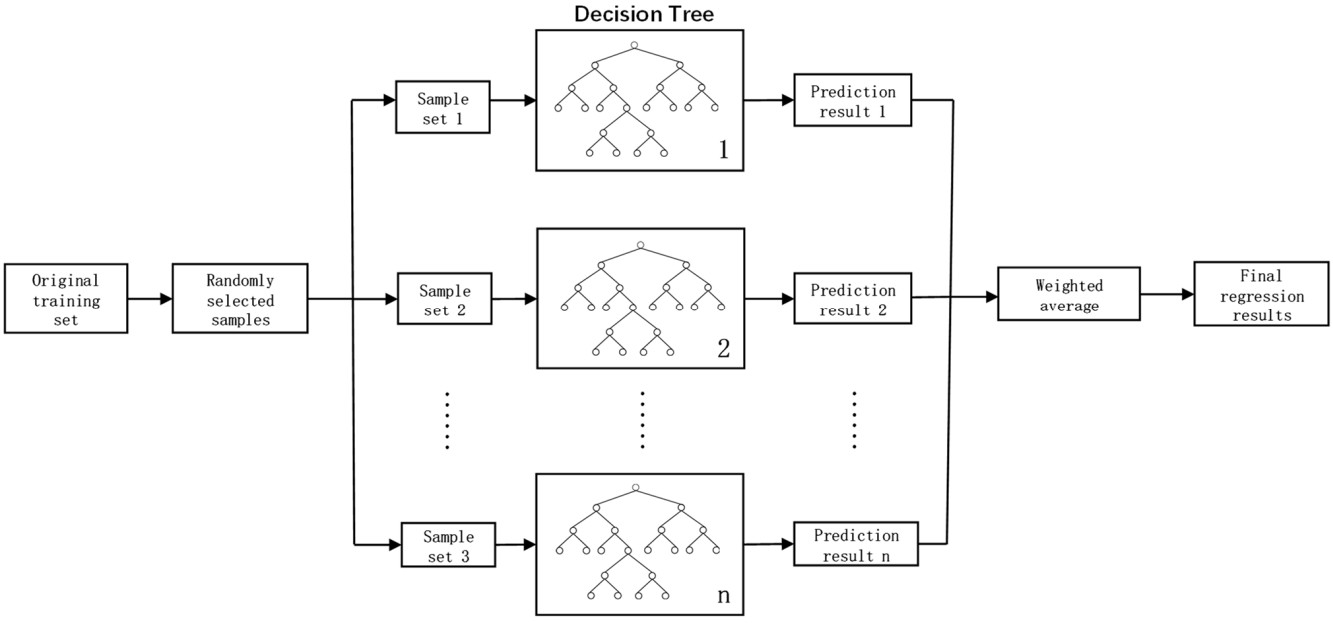

**Figure 1.** Random forest regression model flowchart.

## 3. Results and Analysis

### 3.1. Analysis of the Temporal Dynamics of the Evolution of Emissions and Intensity of Industrial $CO_2$ and LAP

Figure 2 shows the results of studying the dynamic evolution using data on industrial $CO_2$ and LAP emissions, as well as emission intensity in the PRD and YRD urban agglomerations.

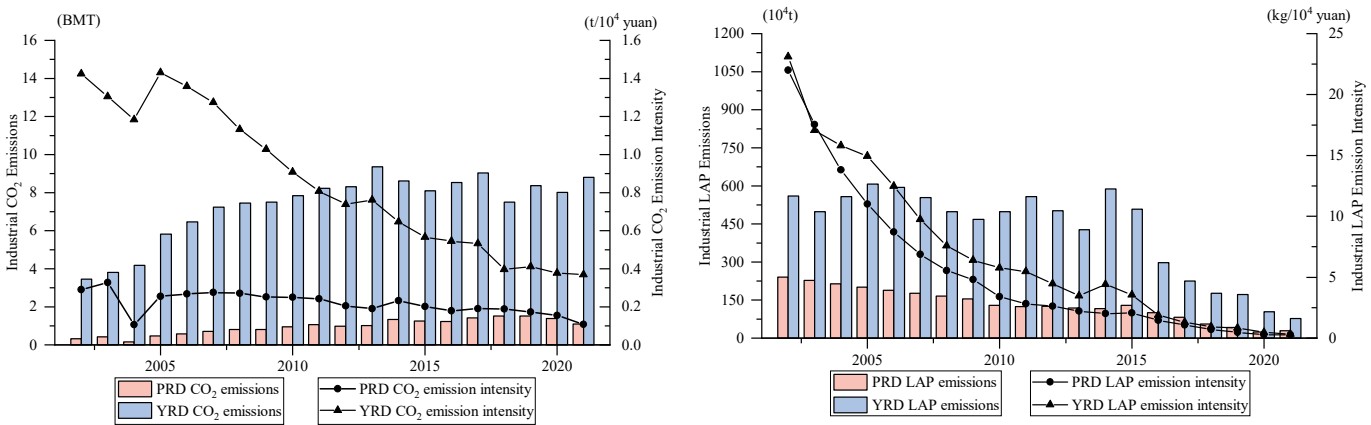

**Figure 2.** Analysis of the time-dynamic evolution of industrial $CO_2$ and LAP emissions and intensities.

Industrial $CO_2$ emissions and emission intensity: Industrial $CO_2$ emissions from the PRD and YRD urban agglomerations are generally increasing, with the YRD urban agglomeration consistently emitting more than the PRD urban agglomeration. The growth rate of PRD urban agglomeration industrial $CO_2$ emissions is more stable compared to YRD urban agglomeration. In the case of YRD, the growth rate of industrial $CO_2$ emissions can be divided into three phases: the first phase from 2002 to 2004 had a flat growth rate, the second phase from 2005 to 2013 had an accelerated growth rate, and the third phase from 2014 to 2021 had a more stable growth rate. Both the PRD and YRD urban agglomerations exhibit a general decrease in industrial $CO_2$ emission intensity. The YRD urban agglomeration shows a larger decrease, indicating a more significant effect of "carbon reduction" in the region during this period. There is still potential for further "carbon reduction".

Industrial LAP emissions and emission intensity: The emissions of industrial LAP and their intensity have decreased in both the PRD and YRD urban agglomerations. However, the YRD urban agglomeration has been leading in terms of industrial LAP emissions, with much higher emissions than the PRD urban agglomeration. The PRD urban agglomeration has shown a stable rate of decline in LAP emissions, while the YRD urban agglomeration has experienced a sudden deceleration in emissions since 2015. Both the PRD and YRD urban agglomerations have shown a decreasing trend in the intensity of industrial LAP emissions. The decreases are significant, indicating that the "pollution reduction" effect of both regions is apparent during this period.

### 3.2. Importance Analysis of Influencing Factors

This paper employs the random forest regression model to analyze the importance of influencing factors on industrial $CO_2$ and LAP emissions. The characteristic screening method is used in the model to reflect the importance of each variable. According to Table 4, an influencing factor is considered the main factor for industrial $CO_2$ and LAP emissions if its characteristic importance is greater than the mean value of 0.125.

**Table 4.** Importance of factors influencing industrial $CO_2$ and LAP emissions.

| Typology | $CO_2$ and LAP Emissions' Common Main Effect Factor | $CO_2$ Emissions' Main Effect Factor | LAP Emissions' Main Effect Factor |
|---|---|---|---|
| Importance of factors influencing | $CO_2$ and LAP > 0.125 | $CO_2$ > 0.125 | LAP > 0.125 |
| Color | | | |

The specific results of this study are shown in Table 5.

**Table 5.** Results on the importance of factors influencing industrial $CO_2$ and LAP emissions.

| Region | | Scale Effect Factors | Structural Effect Factors | Technology Effect Factors | | Governance Effect Factors | | | Trade Effect Factors |
|---|---|---|---|---|---|---|---|---|---|
| | | SC1 | ST1 | TE1 | TE2 | GO1 | GO2 | GO3 | TR1 |
| | Indicators | SC1 | ST1 | TE1 | TE2 | GO1 | GO2 | GO3 | TR1 |
| PRD | LAP | 0.1652 | 0.1216 | 0.1681 | 0.0256 | 0.0227 | 0.1891 | 0.1427 | 0.1650 |
| | $CO_2$ | 0.1780 | 0.0464 | 0.1915 | 0.1104 | 0.0328 | 0.1578 | 0.1498 | 0.1333 |
| | Indicators | SC1 | ST1 | TE1 | TE2 | GO1 | GO2 | GO3 | TR1 |
| YRD | LAP | 0.1724 | 0.1481 | 0.1183 | 0.1862 | 0.0153 | 0.0906 | 0.1712 | 0.0979 |
| | $CO_2$ | 0.1398 | 0.0533 | 0.1507 | 0.1415 | 0.1245 | 0.1169 | 0.1151 | 0.1581 |

It can be concluded that the proportion of industrial value added above the scale (SC1) is the main influence factor of industrial $CO_2$ and LAP emissions; that the proportion of R&D internal expenditure in GDP (TE1) and total trade imports and exports (TR1) are the main influence factors of industrial $CO_2$ emissions; and that the industrial fume and dust removal rate (GO3) is the main driver of industrial LAP emissions.

Industrial value added above the scale (SC1) is the common main influencing factor of industrial $CO_2$ and LAP emissions in the PRD and YRD urban agglomerations. The PRD and YRD urban agglomerations in the south and east of China are two highly developed economic regions. In recent years, their industrial scale has also expanded, and their industrial value added has also increased, leading to an increase in the energy consumption of the two urban agglomerations. This has led to an increase in industrial $CO_2$ and LAP emissions.

The main factors influencing industrial $CO_2$ and LAP emissions are numerous and vary from region to region.

Regarding the structural effect factors, the proportion of industrial value added in GDP (ST1) is the main influencing factor of LAP emissions in the YRD urban agglomeration. Certain differences in industrial structure and economic activity exist between the PRD

and YRD urban agglomerations. High-tech and service industries dominate the PRD urban agglomeration, while manufacturing and heavy industries dominate the YRD urban agglomeration. Therefore, with the increase in energy consumption, the LAP emissions of the YRD urban agglomeration will be affected to a greater extent.

Regarding the technology effect factors, the proportion of R&D internal expenditure in GDP (TE1) is the main influencing factor of $CO_2$ and LAP emissions in the PRD urban agglomeration and $CO_2$ emissions in the YRD urban agglomeration. The industrial structure of YRD urban agglomeration has more secondary industries and fewer tertiary industries, which also leads to the narrowing of the influence of scientific and technological progress due to its industrial structure. Attention should be paid to upgrading and transforming the industrial structure and developing more low-carbon and environmentally friendly industries while not forgetting the treatment of high-polluting industries. Energy consumption per CNY 10,000 of industrial value added (TE2) is the main influencing factor of industrial $CO_2$ and LAP emissions in the YRD urban agglomeration. The YRD urban agglomeration has a higher energy consumption per CNY 10,000 of industrial value added than the PRD urban agglomeration due to its industrial structure. This results in increased industrial $CO_2$ and LAP emissions.

Regarding the governance effect factors, the industrial sulfur dioxide removal rate (GO2) is the main influencing factor of $CO_2$ and LAP emissions in the PRD urban agglomeration. The industrial sulfur dioxide removal rate favors PRD urban agglomeration pollution and carbon reduction efforts, but the effect is insignificant for the YRD urban agglomeration. The removal rate of industrial flue gas (GO3) primarily impacts the emissions of industrial $CO_2$ and LAP in the PRD urban agglomeration, as well as LAP emissions in the YRD urban agglomeration. In the construction of ecological civilization, it is widely recognized that soot and dust are harmful to human health. Relevant laws have set emission standards, the industrial sector has carried out dust removal operations in relevant links, and industrial LAP emissions have decreased. However, it is important to note that while removing dust, the process may consume additional energy due to chemical reactions or equipment usage, which can result in increased industrial $CO_2$ emissions.

Regarding the trade effect factor, the total trade imports and exports (TR1) is the main influencing factor of $CO_2$ and LAP emissions in the PRD urban agglomeration and $CO_2$ emissions in the YRD urban agglomeration. Both PRD and YRD urban agglomerations are important windows for China's foreign trade and account for a large proportion of China's foreign trade. Positive impacts on $CO_2$ reduction can be achieved through the introduction of technology and the environmental constraint requirements of foreign enterprises. At the same time, negative impacts on $CO_2$ reduction can be achieved through pollution transfer and increased energy consumption caused by increased orders.

### 3.3. Response Relationship Analysis of Influencing Factors

To better reveal the action path of the eight influencing factors of industrial $CO_2$ and LAP emissions, this thesis adopts the method of plotting the partial dependence diagram of the influencing factors to indicate the correlation between industrial $CO_2$, LAP emissions, and each influencing factor, as shown in Figures 3 and 4, with which it can be concluded that there is an obvious non-linear relationship between the eight influencing factors and industrial $CO_2$ and LAP emissions.

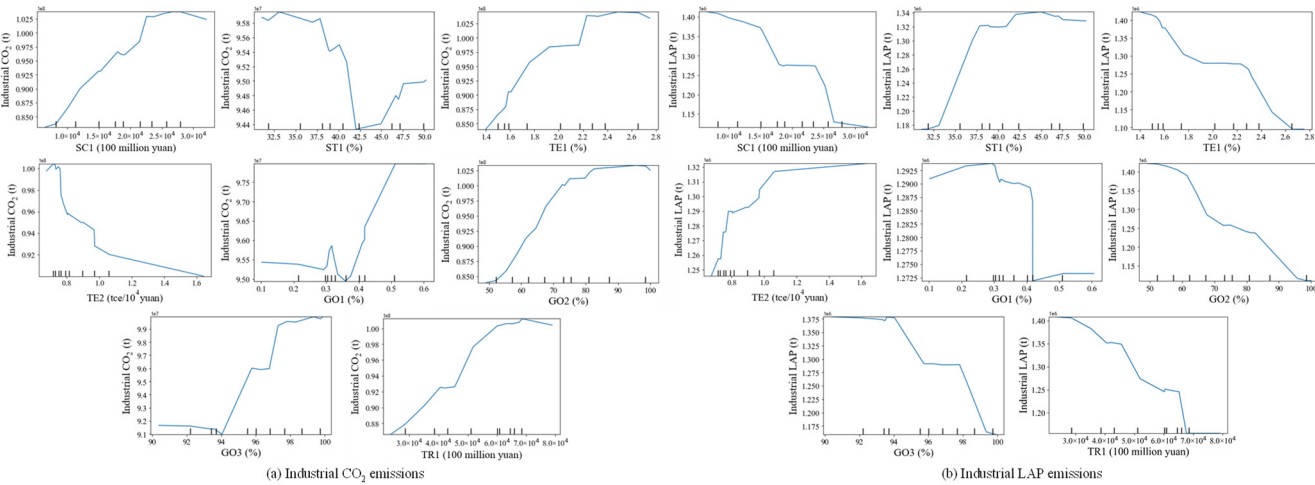

**Figure 3.** Partial dependence diagram of influencing factors of $CO_2$ and LAP emissions in PRD agglomeration.

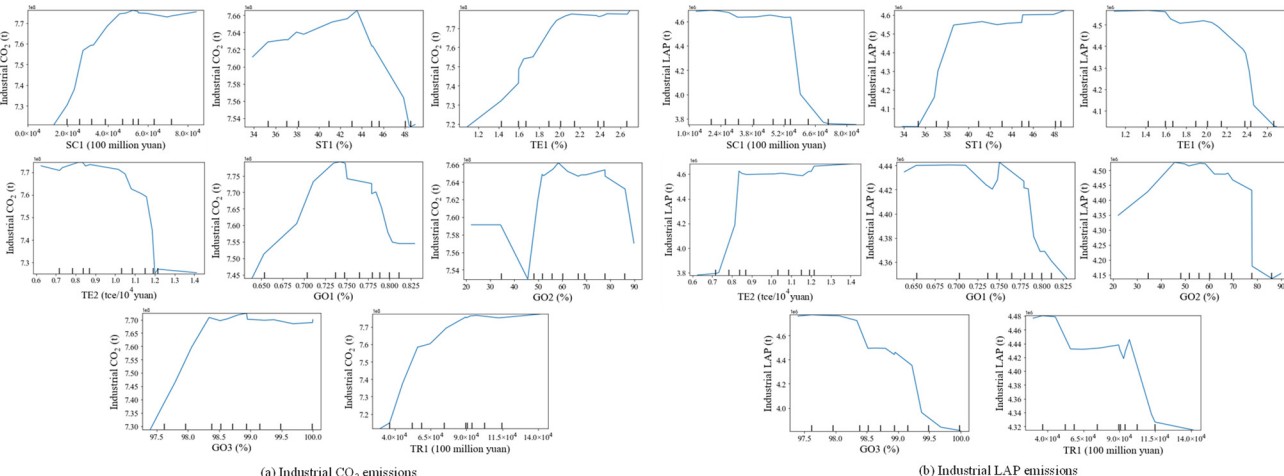

**Figure 4.** Partial dependence diagram of influencing factors of $CO_2$ and LAP emissions in YRD agglomeration.

### 3.3.1. Scale Effect Factors—Industrial Value Added above the Scale (SC1)

Industrial $CO_2$ emissions: The impact of industrial value added above the scale of $CO_2$ emissions in both PRD and YRD urban agglomerations has shown a three-stage upward trend. The trend began with a slow effect, followed by a steep rise in the intermediate stage, and finally reached a steady state in the third stage. The results of both urban agglomerations illustrate that the effect of industrial value added above the scale on $CO_2$ is divided into two stages: within a certain range, the increase in industrial value added above the scale can lead to increased $CO_2$ emissions. However, after exceeding a certain range, the continual increase in industrial value added above the scale suppresses, to a certain extent, and the production activities and the $CO_2$ emissions decline or their growth rate decreases due to the scale effect.

Industrial LAP emissions: The impact of industrial value added above the scale of LAP emissions in both PRD and YRD urban agglomerations showed a three-stage multi-step downward trend, with multiple fluctuating steps in the intermediate stage of development. The impact of industrial value added above the scale in YRD urban agglomeration LAP emissions was weak in the initial stage and declined rapidly after a period of stabilization. The reduction in LAP emissions may be due to enterprises adjusting their industrial structure, particularly by reducing the proportion of pollution-intensive industries. This is an effective method of reducing emissions [26].

### 3.3.2. Structural Effect Factors—The Proportion of Industrial Value Added in GDP (ST1)

Industrial $CO_2$ emissions: The impact of the proportion of industrial value added in GDP of $CO_2$ emissions in the PRD urban agglomeration shows a "V" shape, indicating that the proportion of industrial value added in GDP does not affect $CO_2$ emission reduction. The influence of the proportion of industrial value added in GDP on $CO_2$ emissions in the YRD urban agglomeration shows a three-stage multi-step downward trend from the initial increase in the proportion of industrial value added in GDP; $CO_2$ emissions also increased. In the middle stage, there was a steep decline in $CO_2$ emissions, followed by a steady state in the third stage. When the proportion of industrial value added to GDP is 48.57%, the maximum impact on $CO_2$ emission reduction is achieved. The above indicates that, with the continuous development of industrialization, due to the continuous innovation of technology and society's increasing concern for environmental sustainability, the highly polluting industries at the beginning will be gradually replaced by less polluting industries, which is also one of the reasons why $CO_2$ emissions in the YRD urban agglomeration appear to increase and then decrease as the proportion of industrial value added in GDP increases.

Industrial LAP emissions: The proportion of industrial value added in GDP shows a three-stage stepped upward trend in LAP emissions in both PRD and YRD urban agglomerations, from the initial slow effect to the intermediate stage showing a steep rise and then to the steady state in the third stage. It indicates that with the continuous development of industry and the increase in energy consumption, LAP emissions will also increase.

### 3.3.3. Technology Effect Factors

(1) The proportion of R&D internal expenditure in GDP (TE1)

Industrial $CO_2$ emissions: The proportion of R&D internal expenditure in GDP shows a three-stage upward trend in $CO_2$ emissions in both PRD and YRD urban agglomerations, from a slow effect at the beginning to a steep rise in the middle stage to a steady state in the third stage. "R&D" means research and development. The above results suggest that the proportion of R&D internal expenditure in GDP does not have a significant effect on reducing $CO_2$ emissions, and there may be two reasons for this: Firstly, it is important to note that there may be a threshold for the effect of R&D input on $CO_2$ emissions. If the threshold is lower than a certain level, it can actually cause a rise in $CO_2$ emissions [36]; Secondly, while the investment of scientific research funds has a positive effect on reducing carbon emissions, it may have a lagging effect [37]. It can be observed that during the transformation from lower to higher forms, the effect is not always ideal or significant. However, it is important to pursue long-term effects.

Industrial LAP emissions: The proportion of R&D internal expenditure in GDP shows a three-stage multi-step downward trend in LAP emissions in both PRD and YRD urban agglomerations, from the initial slow effect, the existence of multiple fluctuation steps in the intermediate stage of development, and then the regional steady state in the third stage. This is because the effects of LAP emission reductions will become more and more significant with the investment of scientific research funds and the development of green technology.

(2) Energy consumption per CNY 10,000 of industrial value added (TE2)

Industrial $CO_2$ emissions: Energy consumption per CNY 10,000 of industrial value added shows a trend of increasing, then decreasing, and then leveling off for PRD urban agglomeration $CO_2$ emissions; for YRD urban agglomeration $CO_2$ emissions, they all show a three-stage stepped downward trend, from the initial slow effect to the intermediate stage showing a steep decline, and then a steady state in the third stage. The energy consumption per CNY 10,000 of industrial value added refers to the energy consumption per CNY 10,000 of industrial value added, and the above results show that the energy consumption per CNY 10,000 of industrial value added did not affect the $CO_2$ emission reduction.

Industrial LAP emissions: Energy consumption per CNY 10,000 of industrial value added shows a trend of increasing and then leveling off for LAP emissions in the PRD urban agglomeration; all show a three-stage stepped upward trend, from the initial slow

effect to the intermediate stage showing a steep upward trend, and then to the stable state in the third stage. According to the data analysis, the energy consumption per CNY 10,000 of industrial value added from 2002 to 2021 generally shows a decreasing trend. This indicates that as the energy consumption per CNY 10,000 of industrial value added decreases, the improvement of energy efficiency has a positive impact on the emission reduction of LAP.

3.3.4. Governance Effect Factors

(1) The proportion of environmental investment in GDP (GO1)

Industrial $CO_2$ emissions: The impact of the proportion of environmental protection investment on $CO_2$ emissions in the PRD urban agglomeration shows a three-stage multi-step upward trend, from an initial slow impact, with several fluctuating steps in the intermediate stage of development, to a regional steady state in the third stage; for the YRD urban agglomeration, it shows a trend of increasing, then decreasing, and then leveling off. Some studies show that this is because in dealing with the "three wastes" (i.e., exhaust gas, sewage and industrial waste residues). The relevant departments have adopted an inappropriate treatment method, which leads to a large amount of $CO_2$ emissions when we deal with environmental pollution, resulting in a proportion of the environmental protection investment not having an impact on reducing $CO_2$ emissions. Therefore, we need to make appropriate adjustments to the treatment method. Specifically, it is necessary to adjust the environmental management approach and gradually change the high "carbon emissions" to a low "three wastes" management approach to achieve "low carbon" emissions [38].

Industrial LAP emissions: The share of environmental protection investment in LAP emissions in both PRD and YRD urban agglomerations showed a multi-step downward trend, with several fluctuating steps in the intermediate stage of development and then a regional steady state in the third stage. In particular, the YRD urban agglomeration appeared to have a period of stability in the initial stage, indicating that the increase in the proportion of environmental protection investment, the increase in the amount of funds for ecological civilization construction, the continuous development of green technologies, and the progress of more advanced pollution treatment facilities had a positive impact on the reduction of LAP emissions.

(2) Industrial sulfur dioxide removal rate (GO2) and industrial fume and dust removal rate (GO3)

Industrial $CO_2$ emissions: The industrial sulfur dioxide removal rate showed a three-stage upward trend for $CO_2$ emissions in the PRD urban agglomeration, from an initial slow effect to the intermediate stage where there were multiple fluctuating steps and then to the stable state in the third stage. In the YRD urban agglomeration, the removal rate of industrial sulfur dioxide showed a decreasing trend, followed by an increasing trend, and then another decreasing trend. The rate of industrial fume and dust removal showed a three-stage increasing trend for $CO_2$ emissions in the PRD and YRD urban agglomerations. This trend began with a slow effect, followed by several fluctuating stages in the middle, and finally stabilized in the third stage. This demonstrates that the end-of-pipe treatment of LAP can result in an increase in $CO_2$ emissions. End-of-pipe treatment refers to the process of treating the pollutants generated by the consumption of energy, i.e., the use of desulfurization or dedusting technologies, e.g., in the production of materials for the construction of the LAP system (cement, steel), in the consumption of electricity for the use of the LAP system, and in the chemical reactions during the LAP process, all of which can lead to an increase in $CO_2$ emissions [32]. This indicates a desynchronization between pollution and carbon reduction. The $CO_2$ emissions of the YRD urban agglomeration show a trend of first decreasing, then increasing, and then decreasing with the improvement in the industrial sulfur dioxide removal rate. The reasons for this are as follows: Improving desulphurization efficiency can effectively reduce the emission of harmful gases, such as sulfur dioxide, thereby improving the atmospheric environment and slowing down the formation of acid rain. Secondly, adverse effects are caused by the final processing of LAP, as mentioned previously. Additionally, society as a whole will take appropriate

countermeasures, such as optimizing energy use, improving production processes, and introducing more environmentally friendly technologies, to address the issue of increasing $CO_2$ emissions. The introduction and implementation of a stricter environmental protection system, including the control of carbon emissions from the industry and the energy sector, promotion of the development of renewable energy sources, encouragement of green scientific and technological innovation, and financial incentives for the low-carbon economy, can be implemented with the aim of further reducing $CO_2$ emissions.

Industrial LAP emissions: The industrial sulfur dioxide removal rate and industrial fume and dust removal rate of LAP emissions in the PRD urban agglomeration showed a three-stage multi-step downward trend. There were multiple fluctuation steps in the middle stage of development, followed by regional stability in the third stage. This indicates that the reduction of LAP emissions is affected by the industrial sulfur dioxide removal rate and industrial fume and dust removal rate. The removal rate of industrial sulfur dioxide from the LAP emissions of the YRD urban agglomeration exhibited an "inverted U" trend, suggesting that the reduction of LAP emissions was not affected by the industrial sulfur dioxide removal rate. Similarly, the LAP emissions in the YRD urban agglomeration showed a three-stage multi-step downward trend, with several fluctuation steps in the intermediate stage of development, before reaching the third stage of regional stability. The rate of industrial fume and dust removal affects the reduction of LAP emissions.

### 3.3.5. Trade Effect Factors—Total Trade Imports and Exports (TR1)

Industrial $CO_2$ emissions: Total trade imports and exports show a three-stage upward trend in $CO_2$ emissions in both the PRD and YRD agglomerations, from a slow initial effect to an intermediate stage that shows a steep rise and then to a steady state in the third stage. Total trade imports and exports, i.e., the sum of total trade imports and exports, can be used to observe the overall size of a country in terms of foreign trade. The above shows that as the total scale of foreign trade increases, $CO_2$ emissions increase for two reasons: first, the increase in the total scale of foreign trade itself will promote the production activities of enterprises, which will lead to an increase in energy consumption and hence $CO_2$ emissions; second, some foreign enterprises will transfer pollution-intensive industries to the PRD and YRD urban agglomerations through pollution transfer, which will lead to an increase in energy consumption and hence $CO_2$ emissions [39].

Industrial LAP emissions: Total trade imports and exports show a three-stage downward trend in LAP emissions in both the PRD and YRD agglomerations, from a slow initial effect to a steep decline in the middle stage and a steady state in the third stage. The reason for the decrease in LAP emissions is that with the increase in the overall scale of foreign trade and the introduction of capital and technology from foreign-funded enterprises, the industrial sector can improve its production equipment to meet the requirements of "low energy consumption" and "high efficiency". Some foreign-funded enterprises also choose to trade with enterprises that comply with local environmental technologies or regulations, thus encouraging more enterprises to improve their pollution reduction [27].

## 4. Conclusions and Policy Implications

### 4.1. Conclusions

At this stage, we have achieved the objective of our paper by identifying the factors that affect the need for pollution and carbon reduction in the PRD and YRD urban agglomerations, respectively.

(1) Industrial $CO_2$ emissions in the PRD and YRD urban agglomerations are generally increasing, while emission intensity is generally decreasing, with the YRD urban agglomeration experiencing a greater decline. Both the PRD urban agglomeration and YRD urban agglomeration show a decreasing trend in industrial LAP emissions and intensity in general, and the decrease in industrial LAP emission intensity is larger in both cases. It is evident that pollution reduction and carbon reduction are not synchronized.

While industrial LAP emissions are decreasing, industrial $CO_2$ emissions have not been effectively controlled.

(2) The industrial value added above the scale is the main influencing factor of industrial $CO_2$ and LAP emissions. The proportion of R&D internal expenditure in GDP (TE1) and total trade imports and exports (TR1) are the main influencing factors of industrial $CO_2$ emissions. The industrial fume and dust removal rate (GO3) is the main influencing factor of industrial LAP emissions. The main influencing factors of industrial $CO_2$ and industrial LAP emissions are numerous and have regional differences. It is crucial to maintain a balanced and objective approach when discussing the influencing factors of industrial emissions.

(3) The relationship between influencing factors and industrial $CO_2$ and industrial LAP emissions is non-linear due to the scale effect factor, the lagging nature of R&D expenditure, and the inappropriate treatment of the "three wastes" by relevant departments.

*4.2. Policy Implications*

(1) By analyzing the time-dynamic evolution of industrial $CO_2$ and LAP emissions and intensities, it can be seen that the PRD and YRD urban agglomerations are both more effective in managing industrial LAP. However, the management of industrial $CO_2$ still requires further strengthening. Therefore, both the PRD and YRD urban agglomerations should focus more on the "carbon reduction" perspective in their efforts to reduce carbon emissions. Therefore, the PRD and YRD should prioritize "carbon reduction" and maximize their synergistic effects in reducing industrial $CO_2$ emissions to help achieve the "dual carbon" goal.

(2) Based on the analysis of the importance of factors influencing industrial $CO_2$ and LAP emissions, both the PRD urban agglomeration and YRD urban agglomeration should pay attention to the influence of the scale effect. The industrial sector should first scientifically and efficiently analyze the benefits of economies of scale. Secondly, in the past, many industrial sectors have adopted a careless approach to economic growth, resulting in excessive resource consumption, slow capital turnover, high losses and wastage, low economic efficiency, and so on. We should strive to achieve intensive economic growth.

In the PRD urban agglomeration, the industrial sector should focus on dust and sulfur removal, as well as promoting equipment and technological innovation. The government sector can promote the impact of foreign trade on pollution and carbon reduction by formulating environmental protection standards, engaging in international cooperation, incentivizing the adoption of cleaner technologies, and strengthening regulations to achieve a balance between sustainable economic development and environmental protection.

For the YRD urban agglomeration, the industrial sector should focus on upgrading and transforming the industrial structure, optimizing the allocation of resources among industries, vigorously promoting the strategy of scientific and technological innovations, reducing the number of highly polluting industries, increasing the number of high-tech industries, and reducing energy consumption.

**Author Contributions:** Conceptualization, S.W. and L.C.; methodology, X.W, S.W. and L.C.; software, X.W., Q.Z. and B.Z.; validation, X.W., Q.Z. and B.Z.; formal analysis, X.W., Q.Z. and B.Z.; investigation, X.W., Q.Z. and B.Z.; data curation, X.W., Q.Z. and B.Z.; writing—original draft preparation, X.W.; writing—review and editing, S.W., S.L. and L.C.; visualization, X.W., Q.Z. and B.Z.; supervision, S.W., S.L., L.C. and K.L.; writing and editing, K.L. All authors have read and agreed to the published version of the manuscript.

**Funding:** This work was supported by the Technology Planning Project of Shaoguan, grant numbers 210726224533614 and 210726214533591; the Natural Science Research Project of Shaoguan University, grant numbers SZ2023KJ13; the Philosophy and Social Science Program of Shaoguan, grant number J2020008; the Social Science Program of Shaoguan University, grant number SY2020SK02; the Talent Project of Shaoguan University, grant number 9900064502; the Natural Science Foundation of Guangdong Province, grant numbers 2018A0303100015, 2022A1515011358 and 2023A1515010825;

and the National College Students Innovation and Entrepreneurship Training Program, grant number 202310576013.

**Institutional Review Board Statement:** Not applicable.

**Informed Consent Statement:** Not applicable.

**Data Availability Statement:** The data are not publicly available due to privacy restrictions.

**Conflicts of Interest:** The authors declare no conflicts of interest.

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
