# Peer review of "Impact Factors of Industrial Pollution and Carbon Reduction under the “Dual Carbon” Target: A Case Study of Urban Aggregation in the Pearl River Delta and Yangtze River Delta"

_sustainability, doi:10.3390/su16051879_

Round 1

Reviewer 1 Report

Comments and Suggestions for Authors

The manuscript presents an interesting study. Some comments are as follows.

1. The literature review is not in-depth enough. This study points out the shortcomings of previous studies (lines 88-93) through literature review, but it does not solve these problems well.

2. The innovation of this study is weak. Please highlight the innovation in this study.

3. Why choose the Pearl River Delta and Yangtze River Delta as research objects? What are the similarities and differences between the Pearl River Delta and Yangtze River Delta? Do these regions show inter-provincial heterogeneity? Please add relevant content to the text.

4. The Measurement of industrial carbon dioxide emissions is not comprehensive enough.

5. The number of cites in the YRD urban agglomerations is 27, not 26. Wenzhou is missing.

6. Figure 1 takes up too much space. It can be represented in one picture by different colored lines. The same goes for Figure 2 – Figure 4.

7. In the conclusion section, it is necessary to specify more clearly whether the purpose of the study has been achieved and whether the hypotheses have been confirmed.

Comments on the Quality of English Language

Minor editing of English language required.

Author Response

Response to Reviewer 1’s Comments

Article title: “Impact Factors of Industrial Pollution and Carbon Reduction under the "Dual Carbon" Target :A Case Study of the Pearl River Delta and Yangtze River Delta Urban Aggregation

Review report:

The manuscript presents an interesting study. Some comments are as follows.

  1. The literature review is not in-depth enough. This study points out the shortcomings of previous studies (lines 88-93) through literature review, but it does not solve these problems well.
  2. The innovation of this study is weak. Please highlight the innovation in this study.
  3. Why choose the Pearl River Delta and Yangtze River Delta as research objects? What are the similarities and differences between the Pearl River Delta and Yangtze River Delta? Do these regions show inter-provincial heterogeneity? Please add relevant content to the text.
  4. The Measurement of industrial carbon dioxide emissions is not comprehensive enough.
  5. The number of cities in the YRD urban agglomerations is 27, not 26. Wenzhou is missing.
  6. Figure 1 takes up too much space. It can be represented in one picture by different colored lines. The same goes for Figure 2 – Figure 4.
  7. In the conclusion section, it is necessary to specify more clearly whether the purpose of the study has been achieved and whether the hypotheses have been confirmed.

Response: Thank you very much for your kind and constructive comments on our manuscript. Your comments significantly improved the quality of the manuscript. We have revised the manuscript point by point based on your feedback. Please see the detailed response below.

Point 1: The literature review is not in-depth enough. This study points out the shortcomings of previous studies (lines 88-93) through literature review, but it does not solve these problems well.

Response 1: Thanks for your comments and recommendation. Based on your comments, we consulted more literature, explained and integrated previous research in more depth. For the shortcomings of previous studies, we improved them in our article, highlighted our innovative points, please see line 103-120 in the revised manuscript.

Lines 103-120:

The PRD and YRD urban agglomeration were selected for the study. Up to now, research on the factors influencing pollution and carbon reduction in PRD and YRD urban agglomeration has not been sufficiently explored, and there is insufficient research on inter-provincial heterogeneity to provide management decision support for local governments to formulate differentiated pollution and carbon reduction policies. The PRD and YRD urban agglomeration play a key role in China's economic and social development and simultaneously face unique environmental challenges. This choice considers factors such as concentration of economic activities, different energy mixes, geographical characteristics and differences in potential policy implementation. The two regions are similar regarding high economic development and urbanization, but differ geographic location and industrial composition. In addition, this thesis recognizes the potential inter-provincial heterogeneity of the two city clusters, including differences in environmental policies, resource distribution, and economic status. It aims to provide information on the similarities and differences between the selected city clusters regarding carbon dioxide and LAP emission reductions. By analyzing the data on industrial carbon dioxide (CO2) and industrial local air pollutants (LAP) from 2002 to 2021, the main influencing factors of industrial pollution and carbon reduction are identified, and relevant countermeasure suggestions are given.

Point 2: The innovation of this study is weak. Please highlight the innovation in this study.

Response 2: Thank you very much for your suggestion. Based on your comments, we have highlighted the innovations of this study, please see line 93-120 in the revised manuscript.

Lines 93-120:

Based on this, the thesis systematically studies the influencing factors of pollution and carbon reduction in PRD and YRD urban agglomeration by a using random forest regression model. Through this method, the contribution of multiple variables to pollution and carbon emission reduction can be comprehensively investigated, and the complex relationship between them can be revealed. The study not only focuses on a single factor but also considers multiple potential influencing factors together, providing a more comprehensive and precise reference for formulating environmental protection policies in urban agglomerations. This approach contributes to an in-depth understanding of the interactions of various factors in pollution and carbon reduction. It provides a scientific basis for future sustainable urban development.

The PRD and YRD urban agglomeration were selected for the study. Up to now, research on the factors influencing pollution and carbon reduction in PRD and YRD urban agglomeration has not been sufficiently explored, and there is insufficient research on inter-provincial heterogeneity to provide management decision support for local governments to formulate differentiated pollution and carbon reduction policies. The PRD and YRD urban agglomeration play a key role in China's economic and social development and simultaneously face unique environmental challenges. This choice considers factors such as concentration of economic activities, different energy mixes, geographical characteristics and differences in potential policy implementation. The two regions are similar regarding high economic development and urbanization, but differ geographic location and industrial composition. In addition, this thesis recognizes the potential inter-provincial heterogeneity of the two city clusters, including differences in environmental policies, resource distribution, and economic status. It aims to provide information on the similarities and differences between the selected city clusters regarding carbon dioxide and LAP emission reductions. By analyzing the data on industrial carbon dioxide (CO2) and industrial local air pollutants (LAP) from 2002 to 2021, the main influencing factors of industrial pollution and carbon reduction are identified, and relevant countermeasure suggestions are given.

Point 3: Why choose the Pearl River Delta and Yangtze River Delta as research objects? What are the similarities and differences between the Pearl River Delta and Yangtze River Delta? Do these regions show inter-provincial heterogeneity? Please add relevant content to the text.

Response 3: Thank you for your comment. Detailed revision information is given below. We've added relevant content to the text:

Lines 103-120:

The PRD and YRD urban agglomeration were selected for the study. Up to now, research on the factors influencing pollution and carbon reduction in PRD and YRD urban agglomeration has not been sufficiently explored, and there is insufficient research on inter-provincial heterogeneity to provide management decision support for local governments to formulate differentiated pollution and carbon reduction policies. The PRD and YRD urban agglomeration play a key role in China's economic and social development and simultaneously face unique environmental challenges. This choice considers factors such as concentration of economic activities, different energy mixes, geographical characteristics and differences in potential policy implementation. The two regions are similar regarding high economic development and urbanization, but differ geographic location and industrial composition. In addition, this thesis recognizes the potential inter-provincial heterogeneity of the two city clusters, including differences in environmental policies, resource distribution, and economic status. It aims to provide information on the similarities and differences between the selected city clusters regarding carbon dioxide and LAP emission reductions. By analyzing the data on industrial carbon dioxide (CO2) and industrial local air pollutants (LAP) from 2002 to 2021, the main influencing factors of industrial pollution and carbon reduction are identified, and relevant countermeasure suggestions are given

Point 4: The Measurement of industrial carbon dioxide emissions is not comprehensive enough.

Response 4: Thanks a lot for your comments. Thanks a lot for your comments. Anthropogenic emissions in the thesis refer to carbon dioxide emissions caused by human activities, mainly from the consumption of fossil energy and the burning of biomass fuel. At present, there are three main research methods on carbon dioxide emissions by scholars in China and abroad, namely, Emission-Factor Approach, Mass-Balance Approach and Experiment Approach. Since the research object of this paper is industrial carbon emission, it mainly focuses on fossil energy and adopts the Emission-Factor Approach for calculation. References "Carbon Emission Calculation and Influencing Factor Analysis Based on Industrial Big Data in the "dual-carbon" Era", five raw materials of fossil energy, namely raw coal, gasoline, coke, diesel oil, and natural gas, are selected for calculation.

Lines 133-152:

2.1.1. Measurement of industrial carbon dioxide (CO2) emissions

In this thesis, a total of five fossil energy sources, namely gasoline, diesel oil, raw coal, coke and natural gas, are selected. The CO2 emission from the combustion of the above fossil energy sources is measured by using the emission factor method. On this basis, the carbon emission at the industrial level is measured according to the CO2 emission factors. In academic research, one of the most widely applied carbon emission accounting methods is to convert selected fossil energy sources into standard coal to measure according to a specific carbon emission factor. However, in practice, there are differences in the criteria for determining the carbon emission factor by different organizations and scholars. In this thesis, the coefficients in the Guidelines for the Preparation of Provincial Greenhouse Gas Inventories (Trial) are used to measure carbon emissions, and the formula is as follows [9,10]:

(1)

Where  represents the fuel consumption of the fossil energy source of the ith species.  represents the emission factor of the ith fossil energy source, as shown in Table 1:

Table 1 Carbon dioxide (CO2) emission factors and relevant parameters for fuels

Energy varieties

Average net calorific value 1

Carbon oxidation rate2

Carbon emission factor3

CO2 emission factor4

Gasoline

43070 kJ/m3

0.98

19.1 tc/TJ

2.9251 kJ/m3

Diesel oil

42652 kJ/m3

0.98

20.2 tc/TJ

3.0959 kJ/m3

Raw coal

20908 kJ/m3

0.94

25.8 tc/TJ

1.9003 kJ/m 3

Coke

28435 kJ/m3

0.93

29.2 tc/TJ

2.8604 kJ/m3

Natural gas

38931 kJ/m3

0.99

15.3 tc/TJ

2.1621 kJ/m3

Note: 1) Average net calorific value (NCV) data from the IPCC Guidelines (2006); 2) Carbon oxidation rate is denoted by  expressed; 3) Carbon emission coefficient ( ) data from "Guidelines for Provincial Greenhouse Gas Inventory Preparation (Trial)" (2010); 4) CO2 emission factor:  in kJ/m3.

Point 5: The number of cities in the YRD urban agglomerations is 27, not 26. Wenzhou is missing.

Response 5: We appreciate your question, and thanks a lot for your suggestion. We appreciate your question, and thanks a lot for your suggestion. I would like to offer the following explanation: Wenzhou was only officially stabilized as one of the Yangtze River Delta city groups in 2019, and there was a controversy over the past 20 years as to whether or not Wenzhou belongs to the Yangtze River Delta city group. So, in order to ensure the consistency of data, we have not added the city of Wenzhou at the beginning of the study. Based on your comment, we have corrected the article by adding Wenzhou in the newest data.

Lines 227-232:

2.2. Study area

The PRD and YRD urban agglomeration are taken as research areas in this paper. Located in the south and east of China, these two urban agglomerations are rich in resources and superior geographical advantages, which make them an important engine of China's economic development. The specific cities are shown in Table 3.

Table 2. The study area

Region

Number of cities

Specific cities

the PRD urban agglomerations

9

Guangzhou, Foshan, Zhaoqing, Shenzhen, Dongguan, Huizhou, Zhuhai, Zhongshan, Jiangmen

the YRD urban agglomerations

27

Shanghai, Nanjing, Wuxi, Changzhou, Suzhou, Nantong, Yancheng, Yangzhou, Zhenjiang, Taizhou, Hangzhou, Wenzhou, Ningbo, Jiaxing, Huzhou, Shaoxing, Jinhua, Zhoushan, Taizhou, Hefei, Wuhu, Maanshan, Tongling, Anqing, Chuzhou, Chizhou, Xuancheng;

Point 6: Figure 1 takes up too much space. It can be represented in one picture by different colored lines. The same goes for Figure 2 – Figure 4.

Response 6: Thank you for your recommendation. All indicators are not displayed on one chart because the units of each indicator are inconsistent so they cannot be displayed uniformly. However, we have modified the size of the figures in the revised manuscript as much as possible for clarity. Please refer to lines 250-251, lines 263-264, lines 358-361 of the revised draft for details.

Lines 250-251:

Figure 1 Random forest regression model flowchart

Line 263-264:

Figure 2 Analysis of the time-dynamic evolution of industrial CO2 and LAP emissions

Lines 358-361:

Figure 3 Partial dependence diagram of influencing factors of CO2 and LAP emissions in PRD agglomeration

Figure 4 Partial dependence diagram of influencing factors of CO2 and LAP emissions in YRD agglomeration

Point 7: In the conclusion section, it is necessary to specify more clearly whether the purpose of the study has been achieved and whether the hypotheses have been confirmed.

Response 7: Thank you for the valuable suggestions. In the conclusion section, we have made some revisions. Our research purpose has been achieved by obtaining the main influencing factors of reducing pollution and carbon emissions in the Pearl River Delta and Yangtze River Delta urban agglomerations respectively, so as to obtain countermeasures for two regions to cope with reducing pollution and carbon emissions. The details of the revisions are as follows.

Lines 537-559:

  1. Conclusions and Policy Implications

4.1. Conclusion

At this point, the purpose of our paper has been fulfilled by obtaining the factors influencing the need for pollution and carbon reduction in the PRD and YRD urban agglomeration, respectively.

(1) Industrial CO2 emissions in the PRD and YRD urban agglomeration are generally upward, while emission intensity is generally downward, with the YRD urban agglomeration experiencing a larger decline. Both PRD urban agglomeration and YRD urban agglomeration show a decreasing trend in industrial LAP emissions and intensity in general, and the decrease in industrial LAP emission intensity is larger in both cases. It can be seen that "pollution reduction" and "carbon reduction" are not synchronized, and while industrial LAP emissions are decreasing, industrial CO2 emissions have not been effectively controlled.

(2) The industrial value added above the scale is the common main influencing factor of industrial CO2 and LAP emissions. The proportion of R&D internal expenditure in GDP (TE1) and total trade imports and exports (TR1) are the main influencing factors of industrial CO2 emissions. Industrial fume and dust removal rate (GO3) is the main influencing factor of industrial LAP emissions. The main influencing factors of industrial CO2 and industrial LAP emissions are numerous and have regional differences.

(3) There is a clear non-linear relationship between the influencing factors and industrial CO2 and industrial LAP emissions due to the scale effect factor, the lagging nature of R&D expenditure, and the inappropriate treatment of the "three wastes" by the relevant departments.

Reviewer 2 Report

Comments and Suggestions for Authors

This article represents an empirical study of CO2 and PAL pollution in China's two most prominent industrial regions, the Yangtze River Delta and the Oearl River Delta. The authors employ a complex regression and an enormous dataset to determine which effect factoirs correlate with increases/decreases in pollution levels from 2000-2020. The empirical aspect of the paper is generally strong but the authors are not always successful in explaining their research and its implications. My view is that the paper would be publishable with relatively minor changes, as noted below.

Footnotes--clearly something went wrong here and the AI reader reports missing footnotes, but I suspect the problem may have been formatting. In any case, the footnotes need to be fixed so it is possible to make sense of the authors' references.

Data sources and methods: here the authors could indicate more clearly which ideas they are drawing on from previous studies. Not only are the references not available, the authors should explain which studies they are drawing their "effect factors" from and why they are analyzing them. This is  important for Section 2.1.3 "Indicator System".

Results and Analysis: Figure numbers are repeated. Since there is so much data covered here, it is impressive that the authors have organized and presented their results clearly, for the most part. What are the "three wastes"?

Conclusions and Policy Implications: Here I think the authors could elaborate more on the implications of the research. Where are the correlations strongest and which priorities should local governments pursue first? Industrial producers? In both cases (PRD and YRD), the authros recommend transferring pollution intensive industries, but would that not simply transfer the pollution problem somewhere else (meaning not producing a net carbon reduction as required)? Finally, what does the "constriuction of ecological civilization" mean (last line)?  

Comments on the Quality of English Language

English was generally strong with small errors that should be fixed by a copy editor.

Author Response

Response to Reviewer 2’s Comments

Article title: “Impact Factors of Industrial Pollution and Carbon Reduction under the "Dual Carbon" Target: A Case Study of the Pearl River Delta and Yangtze River Delta Urban Aggregation

Review report:

This article represents an empirical study of CO2 and PAL pollution in China's two most prominent industrial regions, the Yangtze River Delta and the Pearl River Delta. The authors employ a complex regression and an enormous dataset to determine which effect factors correlate with increases/decreases in pollution levels from 2000-2020. The empirical aspect of) the paper is generally strong but the authors are not always successful in explaining their research and its implications. My view is that the paper would be publishable with relatively minor changes, as noted below.

Response: Thank you very much for your recognition and the valuable suggestions! We have addressed the comments point-by-point as below. All the corrections and responses will be incorporated into the new revised manuscript (sustainability-2831921). If further responses and corrections should be made, please don’t hesitate to let me know.

Point 1: Footnotes--clearly something went wrong here and the AI reader reports missing footnotes, but I suspect the problem may have been formatting. In any case, the footnotes need to be fixed so it is possible to make sense of the authors' references.

Response 1: Thanks for your help. We feel really sorry for our carelessness. According to your proposal, we have revised the footnotes of the article. For example, please refer to line 163-183 of the new revised version.

Lines 163-183:

2.1.3. Indicator System

In order to select the main influencing factors of industrial CO2 and LAP reduction in PRD and YRD urban agglomeration, it is usually necessary to consider several aspects. Industrial production activities are an important pathway for industrial CO2 and LAP emissions, so there is an inseparable relationship between their scale and carbon reduction [11]. China's increasing coal-based energy consumption is an important factor in the increase of industrial CO2 and LAP emissions. China's energy consumption is dominated by the industrial sector, which is influenced by high energy consumption, pollution and high carbon emissions, so the optimization of the industrial structure of the industrial sector can have a greater impact on the realization of the reduction of pollution and carbon emission [5]. Technological progress, environmental protection, and management of the increase in investment will have a greater positive effect on reducing pollution and carbon reduction [12]. The impact of import and export on China's carbon emissions is asymmetric, with import trade favoring the reduction of carbon emissions and export trade the opposite [13]. High-quality final products and advanced intermediate products from the import trade can help us improve the technological content of our production activities. At the same time, this has led to a reduction in our country’s carbon emissions [14]. Based on previous research results, combined with the actual situation of PRD and YRD urban agglomeration, this thesis selects a total of eight indicators in five dimensions, namely scale effect factor, structural effect factor, technology effect factor, governance effect factor and trade effect factor, as the explanatory variables of the model.

Point 2: Data sources and methods: here the authors could indicate more clearly which ideas they are drawing on from previous studies. Not only are the references not available, the authors should explain which studies they are drawing their "effect factors" from and why they are analyzing them. This is important for Section 2.1.3 "Indicator System".

Response 2: Thank you very much for your kind and constructive comments on our manuscript. We have provided more powerful evidence for the development of indicators of the article, and the explanation of the index system construction is more comprehensive and effective. We have made the following corrections in the article, as shown in lines 163-183 of the revised draft.

The updated contents are presented below:

Lines 163-183:

2.1.3. Indicator System

In order to select the main influencing factors of industrial CO2 and LAP reduction in PRD and YRD urban agglomeration, it is usually necessary to consider several aspects. Industrial production activities are an important pathway for industrial CO2 and LAP emissions, so there is an inseparable relationship between their scale and carbon reduction [11]. China's increasing coal-based energy consumption is an important factor in the increase of industrial CO2 and LAP emissions. China's energy consumption is dominated by the industrial sector, which is influenced by high energy consumption, pollution and high carbon emissions, so the optimization of the industrial structure of the industrial sector can have a greater impact on the realization of the reduction of pollution and carbon emission [5]. Technological progress, environmental protection, and management of the increase in investment will have a greater positive effect on reducing pollution and carbon reduction [12]. The impact of import and export on China's carbon emissions is asymmetric, with import trade favoring the reduction of carbon emissions and export trade the opposite [13]. High-quality final products and advanced intermediate products from the import trade can help us improve the technological content of our production activities. At the same time, this has led to a reduction in our country’s carbon emissions [14]. Based on previous research results, combined with the actual situation of PRD and YRD urban agglomeration, this thesis selects a total of eight indicators in five dimensions, namely scale effect factor, structural effect factor, technology effect factor, governance effect factor and trade effect factor, as the explanatory variables of the model.

Point 3: Results and Analysis: Figure numbers are repeated. Since there is so much data covered here, it is impressive that the authors have organized and presented their results clearly, for the most part. What are the "three wastes"?

Response 3: Thank you for your reminding. We have corrected the numbers of duplicated charts, as shown in line 253, line 265, line364, line366 of the revised manuscript. We have also updated the term "three wastes" to explain it more clearly, and we have made detailed comments in section 3.3.4.

Lines 453-457: Some studies show that this is because when dealing with the "three wastes" (i.e., waste gas, waste water and industrial waste residue), the relevant departments have adopted an inappropriate treatment method, which leads to a large amount of CO2 emissions when we deal with environmental pollution, resulting in the proportion of environmental protection investment does not have an impact on the reduction of CO2 emissions.

Please refer to lines 250-251, lines 263-264, lines 358-361 of the revised draft for details.

Lines 250-251:

Figure 1 Random forest regression model flowchart

Line 263-264:

Figure 2 Analysis of the time-dynamic evolution of industrial CO2 and LAP emissions

Lines 358-361:

Figure 3 Partial dependence diagram of influencing factors of CO2 and LAP emissions in PRD agglomeration

Figure 4 Partial dependence diagram of influencing factors of CO2 and LAP emissions in YRD agglomeration

Point 4: Conclusions and Policy Implications: Here I think the authors could elaborate more on the implications of the research. Where are the correlations strongest and which priorities should local governments pursue first? Industrial producers? In both cases (PRD and YRD), the author’s recommend transferring pollution intensive industries, but would that not simply transfer the pollution problem somewhere else (meaning not producing a net carbon reduction as required)? Finally, what does the "construction of ecological civilization" mean (last line)?

Response 4: Thank you again for your positive comments and valuable suggestions to improve the quality of our manuscript. We will respond to your suggestions:

  1. The authors have reorganized the conclusions of the study, respectively, for the Pearl River Delta and Yangtze River Delta urban agglomerations, by different sectors, and also to make the conclusions of this paper clearer, thank you again for your suggestions.
  2. I need to explain that it is true that if the transfer of pollution-intensive industries is proposed in both the Pearl River Delta and Yangtze River clusters, there is a risk that the pollution problem will be transferred elsewhere rather than real pollution and carbon reduction, a phenomenon known as "carbon leakage". We need to advocate coordinated global development to achieve sustainable development and environmental protection, and emphasize the need for all countries to share responsibilities and avoid simply transferring environmental problems to other regions. Our original intention is to reduce industry with high pollution and high energy consumption. For this, we are so sorry for the above ill-considered results, and deletes this suggestion in the paper, as shown in lines 560-589 of the revised draft. Thank you again for your valuable advice, and I will pay more attention to rigor in paper writing in the future to avoid the above problems.
  3. When the term "ecological civilization construction" appears in the text, it has already been explained, see lines xxx. We can go further, Ecological civilization construction in China signifies a comprehensive development concept that underscores the harmonious coexistence of humanity and nature. It emphasizes sustainable development and environmental protection as integral components of societal progress. This initiative reflects a commitment to fostering a balanced relationship between economic growth, social well-being, and ecological health. Through a holistic approach, China aims to integrate environmental considerations into all aspects of decision-making and development, promoting a resilient and harmonious coexistence between human activities and the natural environment.

Thank you again for your valuable advice, and I will pay more attention to rigor in future paper writing to avoid the above problems.

Lines 560-589:

4.2. Policy Implications

(1) By analyzing the time-dynamic evolution of industrial CO2 and LAP emissions and intensities, it can be seen that the PRD and YRD urban agglomeration are both more effective in managing industrial LAP. However, management of industrial CO2 still needs to be further strengthened, so both the PRD and YRD urban agglomeration should focus more on the "Carbon Reduction" perspective in their efforts to reduce carbon emissions. Therefore, both the PRD and YRD should focus more on "carbon reduction" and give full play to their synergistic effects in industrial CO2 reduction to help achieve the goal of "dual carbon".

(2) Based on the analysis of the importance of factors influencing industrial CO2 and LAP emissions, both PRD urban agglomeration and YRD urban agglomeration should pay attention to the influence of the scale effect. Firstly, the industrial sector should scientifically and efficiently analyze the benefits of economies of scale; Secondly, in the past, many industrial sectors have adopted a sloppy mode of economic growth, which has led to the high consumption of resources, slow capital turnover, high losses and wastage, and low economic efficiency, and so on. We should strive to achieve intensive economic growth.

For the PRD urban agglomeration, the industrial sector should pay attention to dust and sulphur removal and promote the innovation of equipment and technology. Government departments can promote the impact of foreign trade on pollution and carbon reduction by formulating environmental protection standards, engaging in international co-operation, incentivizing the adoption of cleaner technologies, and strengthening regulation to achieve a balance between sustainable economic development and environmental protection.

For the YRD urban agglomeration, the industrial departments should pay attention to the upgrading and transformation of the industrial structure, optimize the allocation of resources among industries, vigorously promote the strategy of scientific and technological innovation, reduce the number of highly polluting industries, increase the number of high-tech industries and reduce energy consumption. Suggestions for the work of government departments are similar to those for the PRD urban agglomeration.

Reviewer 3 Report

Comments and Suggestions for Authors

1.       Data of CO2 and LPA emission from 2000 to 2020, why not 2022 or 2023? The change of industrial pollution emission is complicated each year. The author needs to update the information in Fig.2 and maunscript.

2.       All references are error. The author needs to double-check.

3.       Format of manuscript needs to double-check (Ex: font, size letter, layout…)

4.       Order of figure is wrong.

5.       Axis of all figures are without unit.

Author Response

Response to Reviewer 3’s Comments

Article title: “Impact Factors of Industrial Pollution and Carbon Reduction under the "Dual Carbon" Target :A Case Study of the Pearl River Delta and Yangtze River Delta Urban Aggregation

Review report:

1.Data of CO2 and LPA emission from 2000 to 2020, why not 2022 or 2023? The change of industrial pollution emission is complicated each year. The author needs to update the information in Fig.2 and manuscript.

2.All references are error. The author needs to double-check.

3.Format of manuscript needs to double-check (Ex: font, size letter, layout…)

4.Order of figure is wrong.

5.Axis of all figures are without unit.

Response: Thank you very much for your kind and constructive comments on our manuscript. Your comments significantly improved the quality of the manuscript. We have revised the manuscript point by point based on your feedback. Please see the detailed response below.

Main Comments:

Point 1: Data of CO2 and LPA emission from 2000 to 2020, why not 2022 or 2023? The change of industrial pollution emission is complicated each year. The author needs to update the information in Fig.2 and manuscript.

Response 1: Thanks a lot for your comments and recommendations. At the time of updating data, it was found that there are still some cities that have not published the 2023 Statistical Yearbook (i.e., 2022 data), and all the cities have published the 2022 Statistical Yearbook (i.e., 2021 data), so in order to ensure data availability and accuracy, we have updated the data to 2021 and corrected it in the article.

Lines 121-132:

  1. Data sources and research methods

2.1. Data sources and variable descriptions

The data of the corresponding indicators for the corresponding regions from 2002 to 2021 were collected through various channels such as China Energy Statistical Yearbook, China Environmental Statistical Yearbook, China Urban Greenhouse Gas Studio Platform, China Economic and Social Big Data Research Platform, and the statistical yearbooks of each province and city of PRD urban agglomeration and YRD urban agglomeration. The data of each indicator and the industrial CO2 and LAP emissions of each city in PRD (9) and YRD urban agglomeration (27) are organized into a dataset, which is convenient for the later model building and data reading.

Point 2: All references are error. The author needs to double-check.

Response 2: Thank you very much for your suggestion. Sorry for the formatting error in the reference. We have updated and corrected the references in the literature. Here are our updates:

Lines 606-668:

References

  1. Chen X, Z.J., Tang X. Synergistic effect of industrial pollution and carbon reduction in China and its influence mechanism. Resource Science 2022, 44, 2387-2398, doi:10.18402/resci.2022.12.01.
  2. Huang R, Z.Q., Wu X et al. Impacts of Economic Structural Transition on Synergetic Control of CO2 and Air Pollutants in Guangdong Province. Research of Environmental Sciences 2022, 35, 2303-2311, doi:10.13198/j.issn.1001-6929.2022.02.30.
  3. Xiang M, W.S., Lv L, et al. Synergistic Paths of Reduced Pollution and Carbon Emissions Based on Different Power Demands in China. Environmental Science 2023, 44, 3637-3648, doi:10.13227/j.hjkx.202207256.
  4. Jiang Y, T.X., Ren K et al. . Research on the drivers of pollution and carbon reduction in China based on double-layer nested SDA. Systems Engineering-Theory & Practice 2022, 44, 3294-3304, doi:10.12011/SETP2022-0408.
  5. Tang X, Z.Y., Cao L, et al. Analysis of the spatial and temporal characteristics of the synergistic effect of pollution and carbon reduction in China and its influence mechanism. Research of Environmental Sciences 2022, 35, 2252-2263, doi:10.13198/j.issn.1001-6929.2022.08.10.
  6. Liu M, L.S., Li J et al. Assessment and prediction of synergistic effect of pollution and carbon reduction in Tianjin. China Environmental 2022, 42, 3940-3949, doi:10.19674/j.cnki.issn1000-6923.20220329.004.
  7. Li Z, X.J., Wang J, et al. Spatial and Temporal Heterogeneity of Urban Carbon Emissions and Their Influencing Factors in Yangtze River Economic Belt. Resources and Environment in the Yangtze Basin 2023, 32, 525-536, doi:10.11870/cjlyzyyhj202303008.
  8. Yuan W, S.H., Wang J,et al. Spatial-temporal Evolution and Driving Forces of Urban Pollution and Carbon Reduction in China. ECONOMIC GEOGRAPHY 2022, 42, doi:10.15957/j.cnki.jjdl.2022.10.009.
  9. Liu W, M.X., Li W et al. Study on decoupling relationship between industrial growth and carbon dioxide emission in the urban agglomeration in the Yellow River Basin. Journal of Environmental Engineering Technology 2023, 13, 849-856, doi:10.12153/j.issn.1674-991X.20220461.
  10. Kang Z, L.W., Liu W. Influencing factors and promoting measures of industrial pollution abatement and carbon reduction of the city clusters in the Yellow River basin. China Environmental Science 2023, 43, 1946-1956, doi:10.19674/j.cnki.issn1000-6923.2023.0046.
  11. Wang F, K.D., Zhu X. Spatial-Temporal Evolution and Driving Force Analysis of Industrial Air Pollutants and Carbon Dioxide Emission Reduction from the Yangtze River Delta. Research of Environmental Sciences 2024, 37, 1-17, doi:10.13198/j.issn.1001-6929.2024.01.08.
  12. Zhang W, X.Y., Hui J. The spatio-temporal impacts and driving factors of the synergistic effects of reducing pollution and carbon emissions in cities of China. Chinese Journal of Environmental Management 2023, 15, 38-47, doi:10.16868/j.cnki.1674-6252.2023.02.038.
  13. Song D, C., Wang B. How Environmental Equity Trading Achieves Pollution Reduction and Carbon Synergies: Theory and Empirical Evidence. Journal of Quantitative & Technological Economics 2022, 1-21, doi:10.13653/j.cnki.jqte.20231214.004.
  14. Guo Q, L.J. Import trade, technology spillover and China’s carbon emissions. China Population, Resources and Environment 2013, 23, 105-109, doi:10.3969/j.issn.1002-2104.2013.03.017.
  15. Fujii H, M.S., Kaneko S. Decomposition analysis of air pollution abatement in China: empirical study for ten industrial sectors from 1998 to 2009. Journal of Cleaner Production 2013, 59, 22-31, doi:10.1016/j.jclepro.2013.06.059.
  16. Lyu W, L.Y., Guan D, et al. . Driving forces of Chinese primary air pollution emissions: an index decomposition analysis. Journal of Cleaner Production 2016, 133, 136-144, doi:10.1016/j.jclepro.2016.04.093.
  17. Li M, W.Q. Study on the Technology Progress Impact on Pollutant Generation Based on Malmquist Index: Take Industry SO2 as an Example. Acta Scientiarum Naturalium Unversitatis Pekinensis 2012, 48, 817-823, doi:10.13209/j.0479-8023.2012.106.
  18. Liu Q, W.Q., Liu Y. Study on the relationship between economic growth, international trade and pollution emissions: an empirical analysis based on SO2 emissions in the United States and China. China Population, Resources and Environment 2012, 22, 170-176, doi:10.3969/j.issn.1002-2104.2012.05.028.
  19. P, Z. End-of-pipe or process-integrated: evidence from LMDI decomposition of China's SO2 emission density reduction. FRONTIERS OF ENVIRONMENTAL SCIENCE & ENGINEERING 2013, 7, 867-874, doi:10.1007/s11783-013-0541-0.
  20. Cui D, J.B. Comprehensive evaluation of water ecological civilization based on random forests regression algorithm. Advances in Science and Technology of Water Resources 2014, 56-60,79, doi:10.3880 /j.issn.1006-7647.2014.05.011.
  21. Zhao Long, S.G., Wu Wei et al. Rainfall Prediction Model for Mountain Torrent Disaster Based on Random Forest Regression Algorithm. Journal of University of Jinan (Science and Technology) 2022, 36, 404-411,423, doi:10.13349/j.cnki.jdxbn.20220325.001.
  22. Cheng Y, X.C., Ren J et al. Atmospheric Environment Effect of Industrial Structure Evolution in Shandong Province. CHINA POPULATION,RESOURCES AND ENVIRONMENT 2014, 24, 157-162, doi:10.3969/j.issn.1002-2104.2014.01.022.
  23. Bian Y, L.X., Zhou X,et al. Temporal and spatial evolution characteristics and influencing factors of industrial carbon emissions in Beijing-Tianjin-Hebei region. Environmental Science & Technology 2021, 44, 37-47, doi:10.19672/j.cnki.1003-6504.1329.21.338.
  24. Wang Z, W.L. A Study on the Relationship Among R&D Input, Upgrading Industrial Structure and Carbon Emission. Journal of Industrial Technological Economics 2019, 38, 62-70, doi:10.3969/j.issn.1004-910X.2019.05.008.
  25. Yu W, Z.T., Shen D. County-levelspatial pattern and influencing factors evolution of carbon emission intensity in China: A random forest model analysis. China Environmental Science 2022, 42, 2788-2798, doi:10.19674/j.cnki.issn1000-6923.20220219.001.
  26. Xu K, W.Y. The Influence of China’s OFDI on Its Domestic CO2 Emissions: An Empirical Analysis Based on China’s Provincial Panel Data. GUOJI SHANGWU YANJIU 2015, 76-86, doi:10.13680/j.cnki.ibr.2015.01.008.
  27. Duan J, Y.J. Wet desulphurisation and carbon emission. Thermal Power Generation 2011, 40, 83-84.
  28. Chen Y, L.K. Factor decomposition of SO2 emission intensity in Chinese industrial sectors and its influencing factors--an analysis based on the backward and forward linkage of FDI industries. Management World 2010, 14-21, doi:10.19744/j.cnki.11-1235/f.2010.03.003.

Point 3: Format of manuscript needs to double-check (Ex: font, size letter, layout…)

Response 3: Thank you for your comment. We have revised the format of the manuscript (font style, font size, chart format, etc.), and we will pay more attention to the format, grammar and other details of English writing, and strive to increase the readability of the paper.

Point 4: Order of figure is wrong.

Response 4: Thanks a lot for your comments. Sorry for our carelessness. We have modified the numerical order of the chart. Please refer to lines 250-251, lines 263-264, lines 358-361 of the revised draft for details.

Lines 250-251:

Figure 1 Random forest regression model flowchart

Line 263-264:

Figure 2 Analysis of the time-dynamic evolution of industrial CO2 and LAP emissions

Lines 358-361:

Figure 3 Partial dependence diagram of influencing factors of CO2 and LAP emissions in PRD agglomeration

Figure 4 Partial dependence diagram of influencing factors of CO2 and LAP emissions in YRD agglomeration

Point 5: Axis of all figures are without unit.

Response 5: We appreciate your question. The reason why our figures are without units is because our diagrams are Partial Dependency Plots, and the sample plots in the scikit-learn also doesn’t have units.

But you gave us good suggestions to help us improve the readability of the paper. We have updated the charts as much as possible, to make it easier to read and understand.

Lines 358-361:

Figure 3 Partial dependence diagram of influencing factors of CO2 and LAP emissions in PRD agglomeration

Figure 4 Partial dependence diagram of influencing factors of CO2 and LAP emissions in YRD agglomeration

Reviewer 4 Report

Comments and Suggestions for Authors

Energy and environmental issues have become important factors in the global economic and social development. The study is interesting and meaningful. However, there were some issues that should be improved in this manuscript.

1. In the Introduction section, the "Double Carbon" policies of other major countries should be briefly introduced.

2. The references on pollution reduction and carbon reduction are insufficient, some references should be cited and analyzed, e.g., DOI: 10.1016/j.fuel.2023.129925; 10.1016/j.mineng.2023.108296

3. There are some errors, e.g. Lines 328, 354, 356, 394, 415, “Error! Reference source not found.”; Line 64, “national level[1] local level”; etc.

4. The order of figures are wrong. It is suggested that Figures 1-4 (Pages 9-12) should be merged.

5. The titles of Sections 3.3.1-3.3.5 should be revised.

6. In Table 4, the number of digits after the decimal point should be consistent.

7. There is too much content in Section 4.2. Policy Implications and should be simplified.

8. There are some syntax errors, sentence errors and other English Language errors that must be corrected, e.g. “for example, Tang Xiangbo et al.[6] For example, Tang Xiangbo et al.”; Line 111, “In this thesis”. Possibly, a native English Language speaking Scientists should be employed for the final editing.

9. The format of references should be consistent, e.g. Line 443, 464, 490, 516, 522, 533, 544, the page number is missed. Line 548, “Huang Ruxia, Zhong Qiumeng, Wu Xiaohui”; Line 558, “Li Z.J., Xu J.L., Wang J”; Line 596, “FRONTIERS OF ENVIRONMENTAL SCIENCE & ENGINEERING”; etc.

Comments on the Quality of English Language

Moderate editing of English language required.

Author Response

Response to Reviewer 4’s Comments

Article title: “Impact Factors of Industrial Pollution and Carbon Reduction under the "Dual Carbon" Target :A Case Study of the Pearl River Delta and Yangtze River Delta Urban Aggregation

Review report:

Energy and environmental issues have become important factors in the global economic and social development. The study is interesting and meaningful. However, there were some issues that should be improved in this manuscript.

  1. In the Introduction section, the "Double Carbon" policies of other major countries should be briefly introduced.
  2. The references on pollution reduction and carbon reduction are insufficient, some references should be cited and analyzed, e.g., DOI: 10.1016/j.fuel.2023.129925; 10.1016/j.mineng.2023.108296.
  3. There are some errors, e.g. Lines 328, 354, 356, 394, 415, “Error! Reference source not found.”; Line 64, “national level[1] local level”; etc.
  4. The order of figures are wrong. It is suggested that Figures 1-4 (Pages 9-12) should be merged.
  5. The titles of Sections 3.3.1-3.3.5 should be revised.
  6. In Table 4, the number of digits after the decimal point should be consistent.
  7. There is too much content in Section 4.2. Policy Implications and should be simplified.
  8. There are some syntax errors, sentence errors and other English Language errors that must be corrected, e.g. “for example, Tang Xiangbo et al.[6] For example, Tang Xiangbo et al.”; Line 111, “In this thesis”. Possibly, a native English Language speaking Scientists should be employed for the final editing.
  9. The format of references should be consistent, e.g. Line 443, 464, 490, 516, 522, 533, 544, the page number is missed. Line 548, “Huang Ruxia, Zhong Qiumeng, Wu Xiaohui”; Line 558, “Li Z.J., Xu J.L., Wang J”; Line 596, “FRONTIERS OF ENVIRONMENTAL SCIENCE & ENGINEERING”; etc.

Response: Thank you very much for your kind and constructive comments on our manuscript. Your comments significantly improved the quality of the manuscript. We have revised the manuscript point by point based on your feedback. Please see the detailed response below.

Main Comments:

Point 1: In the Introduction section, the "Double Carbon" policies of other major countries should be briefly introduced.

Response 1: We have added a new small section to the article for a brief overview of " dual-carbon " policies in other major countries. Here is the section we have modified:

Lines 46-63:

The "dual-carbon" goal aims to promote China's high-quality development and the construction of ecological civilization (a comprehensive development concept in China that emphasises the harmonious coexistence of human beings with nature and focuses on sustainable development and environmental protection). In addition, countries worldwide are also adopting different "dual-carbon" policies aimed at mitigating climate change and achieving carbon peak and neutrality. For example, in the United States, "dual-carbon" policies include re-joining the Paris Agreement, setting ambitious carbon-neutral goal in 2050, and investing heavily in clean energy and infrastructure. Concerns about specific sector objectives, policy reversals and cooperation with domestic and international partners have formed the policy of this administration, emphasizing comprehensive climate action. The European Union's ambitious "Green Deal" aims to transform its member States into carbon-neutral economies by 2050, addressing the energy transition, the promotion of renewable energy and industrial innovation. Despite its energy-intensive industries, India is aggressively pursuing carbon reduction targets and investing in renewable energy and clean technologies for sustainable development. These policies reflect the consensus of global cooperation in tackling climate change and highlight the differences and challenges of countries in achieving carbon neutrality

Point 2: The references on pollution reduction and carbon reduction are insufficient, some references should be cited and analyzed, e.g., DOI: 10.1016/j.fuel.2023.129925; 10.1016/j.mineng.2023.108296.

Response 2: Thanks a lot for your comments and recommendations. We have cited more references based on your kind suggestion and made more analysis based on previous references. Specific corrections are as follows:

Lines 606-668:

References

  1. Chen X, Z.J., Tang X. Synergistic effect of industrial pollution and carbon reduction in China and its influence mechanism. Resource Science 2022, 44, 2387-2398, doi:10.18402/resci.2022.12.01.
  2. Huang R, Z.Q., Wu X et al. Impacts of Economic Structural Transition on Synergetic Control of CO2 and Air Pollutants in Guangdong Province. Research of Environmental Sciences 2022, 35, 2303-2311, doi: 10.13198/j.issn.1001-6929.2022.02.30.
  3. Xiang M, W.S., Lv L, et al. Synergistic Paths of Reduced Pollution and Carbon Emissions Based on Different Power Demands in China. Environmental Science 2023, 44, 3637-3648, doi: 10.13227/j.hjkx.202207256.
  4. Jiang Y, T.X., Ren K et al. Research on the drivers of pollution and carbon reduction in China based on double-layer nested SDA. Systems Engineering-Theory & Practice 2022, 44, 3294-3304, doi:10.12011/SETP2022-0408.
  5. Tang X, Z.Y., Cao L, et al. Analysis of the spatial and temporal characteristics of the synergistic effect of pollution and carbon reduction in China and its influence mechanism. Research of Environmental Sciences 2022, 35, 2252-2263, doi: 10.13198/j.issn.1001-6929.2022.08.10.
  6. Liu M, L.S., Li J et al. Assessment and prediction of synergistic effect of pollution and carbon reduction in Tianjin. China Environmental 2022, 42, 3940-3949, doi: 10.19674/j.cnki.issn1000-6923.20220329.004.
  7. Li Z, X.J., Wang J, et al. Spatial and Temporal Heterogeneity of Urban Carbon Emissions and Their Influencing Factors in Yangtze River Economic Belt. Resources and Environment in the Yangtze Basin 2023, 32, 525-536, doi:10.11870/cjlyzyyhj202303008.
  8. Yuan W, S.H., Wang J, et al. Spatial-temporal Evolution and Driving Forces of Urban Pollution and Carbon Reduction in China. ECONOMIC GEOGRAPHY 2022, 42, doi: 10.15957/j.cnki.jjdl.2022.10.009.
  9. Liu W, M.X., Li W et al. Study on decoupling relationship between industrial growth and carbon dioxide emission in the urban agglomeration in the Yellow River Basin. Journal of Environmental Engineering Technology 2023, 13, 849-856, doi: 10.12153/j.issn.1674-991X.20220461.
  10. Kang Z, L.W., Liu W. Influencing factors and promoting measures of industrial pollution abatement and carbon reduction of the city clusters in the Yellow River basin. China Environmental Science 2023, 43, 1946-1956, doi: 10.19674/j.cnki.issn1000-6923.2023.0046.
  11. Wang F, K.D., Zhu X. Spatial-Temporal Evolution and Driving Force Analysis of Industrial Air Pollutants and Carbon Dioxide Emission Reduction from the Yangtze River Delta. Research of Environmental Sciences 2024, 37, 1-17, doi: 10.13198/j.issn.1001-6929.2024.01.08.
  12. Zhang W, X.Y., Hui J. The spatio-temporal impacts and driving factors of the synergistic effects of reducing pollution and carbon emissions in cities of China. Chinese Journal of Environmental Management 2023, 15, 38-47, doi: 10.16868/j.cnki.1674-6252.2023.02.038.
  13. Song D, C., Wang B. How Environmental Equity Trading Achieves Pollution Reduction and Carbon Synergies: Theory and Empirical Evidence. Journal of Quantitative & Technological Economics 2022, 1-21, doi: 10.13653/j.cnki.jqte.20231214.004.
  14. Guo Q, L.J. Import trade, technology spillover and China’s carbon emissions. China Population, Resources and Environment 2013, 23, 105-109, doi: 10.3969/j.issn.1002-2104.2013.03.017.
  15. Fujii H, M.S., Kaneko S. Decomposition analysis of air pollution abatement in China: empirical study for ten industrial sectors from 1998 to 2009. Journal of Cleaner Production 2013, 59, 22-31, doi: 10.1016/j.jclepro.2013.06.059.
  16. Lyu W, L.Y., Guan D, et al. Driving forces of Chinese primary air pollution emissions: an index decomposition analysis. Journal of Cleaner Production 2016, 133, 136-144, doi: 10.1016/j.jclepro.2016.04.093.
  17. Li M, W.Q. Study on the Technology Progress Impact on Pollutant Generation Based on Malmquist Index: Take Industry SO2 as an Example. Acta Scientiarum Naturalium Unversitatis Pekinensis 2012, 48, 817-823, doi:10.13209/j.0479-8023.2012.106.
  18. Liu Q, W.Q., Liu Y. Study on the relationship between economic growth, international trade and pollution emissions: an empirical analysis based on SO2 emissions in the United States and China. China Population, Resources and Environment 2012, 22, 170-176, doi: 10.3969/j.issn.1002-2104.2012.05.028.
  19. P, Z. End-of-pipe or process-integrated: evidence from LMDI decomposition of China's SO2 emission density reduction. FRONTIERS OF ENVIRONMENTAL SCIENCE & ENGINEERING 2013, 7, 867-874, doi:10.1007/s11783-013-0541-0.
  20. Cui D, J.B. Comprehensive evaluation of water ecological civilization based on random forests regression algorithm. Advances in Science and Technology of Water Resources 2014, 56-60,79, doi:10.3880 /j. issn.1006-7647.2014.05.011.
  21. Zhao Long, S.G., Wu Wei et al. Rainfall Prediction Model for Mountain Torrent Disaster Based on Random Forest Regression Algorithm. Journal of University of Jinan (Science and Technology) 2022, 36, 404-411,423, doi: 10.13349/j.cnki.jdxbn.20220325.001.
  22. Cheng Y, X.C., Ren J et al. Atmospheric Environment Effect of Industrial Structure Evolution in Shandong Province. CHINA POPULATION, RESOURCES AND ENVIRONMENT 2014, 24, 157-162, doi: 10.3969/j.issn.1002-2104.2014.01.022.
  23. Bian Y, L.X., Zhou X, et al. Temporal and spatial evolution characteristics and influencing factors of industrial carbon emissions in Beijing-Tianjin-Hebei region. Environmental Science & Technology 2021, 44, 37-47, doi: 10.19672/j.cnki.1003-6504.1329.21.338.
  24. Wang Z, W.L. A Study on the Relationship Among R&D Input, Upgrading Industrial Structure and Carbon Emission. Journal of Industrial Technological Economics 2019, 38, 62-70, doi: 10.3969/j.issn.1004-910X.2019.05.008.
  25. Yu W, Z.T., Shen D. County-levelspatial pattern and influencing factors evolution of carbon emission intensity in China: A random forest model analysis. China Environmental Science 2022, 42, 2788-2798, doi: 10.19674/j.cnki.issn1000-6923.20220219.001.
  26. Xu K, W.Y. The Influence of China’s OFDI on Its Domestic CO2 Emissions: An Empirical Analysis Based on China’s Provincial Panel Data. GUOJI SHANGWU YANJIU 2015, 76-86, doi: 10.13680/j.cnki.ibr.2015.01.008.
  27. Duan J, Y.J. Wet desulphurisation and carbon emission. Thermal Power Generation 2011, 40, 83-84.
  28. Chen Y, L.K. Factor decomposition of SO2 emission intensity in Chinese industrial sectors and its influencing factors--an analysis based on the backward and forward linkage of FDI industries. Management World 2010, 14-21, doi: 10.19744/j.cnki.11-1235/f.2010.03.003.

Point 3: There are some errors, e.g. Lines 328, 354, 356, 394, 415, “Error! Reference source not found.”; Line 64, “national level [1] local level”; etc.

Response 3: Thanks a lot for your comments. We apologize for the error in this area. We have corrected it. All references have corresponding footnotes.

Point 4: The order of figures are wrong. It is suggested that Figures 1-4 (Pages 9-12) should be merged.

Response 4: Thanks a lot for your comments and recommendations. We have modified the numerical order of the charts and have a more logical layout; we have laid out the specific charts as follows. Please refer to lines 250-251, lines 263-264, lines 358-361 of the revised draft for details.

Lines 250-251:

Figure 1 Random forest regression model flowchart

Line 263-264:

Figure 2 Analysis of the time-dynamic evolution of industrial CO2 and LAP emissions

Lines 358-361:

Figure 3 Partial dependence diagram of influencing factors of CO2 and LAP emissions in PRD agglomeration

Figure 4 Partial dependence diagram of influencing factors of CO2 and LAP emissions in YRD agglomeration

Point 5: The titles of Sections 3.3.1-3.3.5 should be revised.

Response 5: Thank you for your feedback and recommendations. We have made corrections to the title and we will pay more attention to the formatting, grammar of English writing. Here are the corrections we have made:

Lines 362-536:

3.3.1. Scale effect factors-Industrial value added above the scale (SC1)

Industrial CO2 emissions: The impact of industrial value added above the scale of CO2 emissions in both PRD and YRD urban agglomeration shows a three-stage upward trend, from the initial slow effect to the intermediate stage, which shows a steep rise and then to the steady state in the third stage. The results of both urban agglomerations illustrate that the effect of industrial value added above the scale on CO2 is divided into two stages: within a certain range, the increase in industrial value added above the scale can lead to an increase in CO2 emissions. However, after exceeding a certain range, the continual increase in industrial value added above the scale suppresses, to a certain extent, the production activities and the CO2 emissions decline or their growth rate decreases due to the scale effect.

Industrial LAP emissions: The impact of industrial value added above the scale of LAP emissions in both PRD and YRD urban agglomeration showed a three-stage multi-step downward trend, with multiple fluctuating steps in the intermediate stage of development. The impact of industrial value added above the scale in YRD urban agglomeration LAP emissions was weak in the initial stage and declined rapidly after a period of stabilization; This may be due to the adjustment of the industrial structure by enterprises, in particular, to reduce the proportion of pollution-intensive industries, which is an important way of reducing LAP emissions [22].

3.3.2. Structural effect factors-The proportion of industrial value added in GDP(ST1)

Industrial CO2 emissions: The impact of the proportion of industrial value added in GDP of CO2 emissions in the PRD urban agglomeration shows a "V" shape, indicating that the proportion of industrial value added in GDP does not affect CO2 emission reduction. The influence of the proportion of industrial value added in GDP on CO2 emissions in the YRD urban agglomeration shows a three-stage multi-step downward trend, from the initial increase in the proportion of industrial value added in GDP, CO2 emissions also increased. To the middle stage of the steep decline in the situation, and then to the third stage of the steady state, and when the proportion of industrial value added in , its impact on the CO2 emission reduction reached the maximum. The above indicates that, with the continuous development of industrialization, due to the continuous innovation of technology and society's increasing concern for environmental sustainability, the highly polluting industries at the beginning will be gradually replaced by less polluting industries [23], which is also one of the reasons why CO2 emissions in the YRD urban agglomeration appear to increase and then decrease as the proportion of industrial value added in GDP increases.

Industrial LAP emissions: The proportion of industrial value added in GDP shows a three-stage stepped upward trend in LAP emissions in both PRD and YRD urban agglomeration, from the initial slow effect to the intermediate stage showing a steep rise and then to the steady state in the third stage. It indicates that with the continuous development of industry and the increase in energy consumption, LAP emissions will also increase.

3.3.3. Technology effect factors

(1) The proportion of R&D internal expenditure in GDP(TE1)

Industrial CO2 emissions: The proportion of R&D internal expenditure in GDP shows a three-stage upward trend in CO2 emissions in both PRD and YRD urban agglomeration, from a slow effect at the beginning to a steep rise in the middle stage and then to a steady state in the third stage. "R&D" means research and development; the above results indicate that the proportion of R&D internal expenditure in GDP does not have an emission reduction effect on CO2 emissions, and there may be two reasons for this: firstly, because there may be a threshold for the effect of R&D input on CO2 emissions, and lower than a certain threshold will cause a rise in CO2 emissions[24]; And secondly, although the investment of scientific research funds has a positive effect on carbon emission reduction, it has a lagging effect[25], and it can be seen in the process of transformation from lower forms to higher forms that the effect is not ideal or significant. In lower forms to higher forms of the conversion process, the effect can be seen in the short term as not ideal or not significant, and there is a need to pursue long-term effects.

Industrial LAP emissions: The proportion of R&D internal expenditure in GDP shows a three-stage multi-step downward trend in LAP emissions in both PRD and YRD urban agglomeration, from the initial slow effect, the existence of multiple fluctuation steps in the intermediate stage of development, and then the regional steady state in the third stage. This is because the effect of LAP emission reduction will become more and more significant with the investment of scientific research funds and the development of green technology.

(2) Energy consumption per 10,000 yuan of industrial value added (TE2)

Industrial CO2 emissions: Energy consumption per 10,000 yuan of industrial value added shows a trend of rising and then falling and then leveling off for PRD urban agglomeration CO2 emissions, and for YRD urban agglomeration CO2 emissions, all show a three-stage stepped downward trend, from the initial slow effect to the intermediate stage showing a steep decline, and then to the third stage of a steady state. The energy consumption per 10,000 yuan of industrial value added refers to the energy consumption per 10,000 yuan of industrial value added, and the above results show that the energy consumption per 10,000 yuan of industrial value added did not affect on CO2 emission reduction.

Industrial LAP emissions: Energy consumption per 10,000 yuan of industrial value added shows a trend of rising and then leveling off for PRD urban agglomeration LAP emissions, all show a three-stage stepped upward trend, from the initial slow effect to the intermediate stage showing a steep upward trend, and then to the stable state in the third stage. According to the data analysis, it can be obtained that the energy consumption per 10,000 yuan of industrial value added from 2002 to 2021 shows a decreasing trend in general. This indicates that as the energy consumption per 10,000 yuan of industrial value-added decreases, i.e., the improvement of energy utilization has a positive effect on the emission reduction of LAP.

3.3.4. Governance effect factors

(1) The proportion of environmental investment in GDP(GO1)

Industrial CO2 emissions: The impact of the proportion of environmental protection investment on CO2 emissions in the PRD urban agglomeration shows a three-stage, multi-step upward trend, from an initial slow impact, with multiple fluctuating steps in the intermediate stage of development, to a regional steady state in the third stage; And to the YRD urban agglomeration, a trend of rising and then declining and then levelling off. Some studies show that this is because when dealing with the "three wastes" (i.e., waste gas, waste water and industrial waste residue), the relevant departments have adopted an inappropriate treatment method, which leads to a large amount of CO2 emissions when we deal with environmental pollution, resulting in the proportion of environmental protection investment does not have an impact on the reduction of CO2 emissions. Therefore, we need to make corresponding adjustments to the treatment method. Specifically, it is necessary to adjust the environmental management approach and gradually change the high "carbon emissions" to a low "three wastes" management approach to achieve the real "low-carbon"[26].

Industrial LAP emissions: The proportion of environmental protection investment on LAP emissions in both PRD and YRD urban agglomeration showed a multi-step downward trend, with multiple fluctuating steps in the intermediate stage of development and then a regional steady state in the third stage. In particular, YRD urban agglomeration appeared a period of stability in the initial stage, which indicates that the increase in the proportion of investment in environmental protection, the increase in the amount of funds for ecological civilization construction, the continuous development of green technologies and the progress of more advanced pollution treatment facilities had a positive impact on LAP emission reduction.

(2) Industrial sulfur dioxide removal rate (GO2) & Industrial fume and dust removal     rate (GO3)

Industrial CO2 emissions: The industrial sulfur dioxide removal rate showed a three-stage upward trend in CO2 emissions in the PRD urban agglomeration, from the initial slow effect to the intermediate stage, where there are multiple fluctuation steps, and then to the stable state in the third stage; The industrial sulfur dioxide removal rate showed a decreasing, then increasing, and then decreasing trend in YRD urban agglomeration. The industrial fume and dust removal rate showed a three-step upward trend for both PRD and YRD urban agglomeration CO2 emissions, from the initial slow effect to multiple fluctuating steps in the intermediate stage and then to a steady state in the third stage. The above illustrates that end-of-pipe treatment of air pollutant LAP may lead to an increase in CO2 emissions. End-of-pipe treatment refers to the process of treating the pollutants produced by the consumption of energy, i.e., the use of desulphurization or de-dusting technologies, e.g., in the production of materials for the construction of the LAP system (cement, steel), in the consumption of electricity for the use of the LAP system, and in the chemical reactions during the LAP process, all of which can lead to an increase in CO2 emissions [27]. This also suggests that there is a desynchronization between pollution and carbon reduction. With the improvement of industrial sulfur dioxide removal rate, the CO2 emission of YRD urban agglomeration shows a trend of first decreasing, then increasing, and then decreasing. The reasons are as follows: First, the improvement of desulfurization efficiency can effectively reduce the emission of harmful gases such as sulfur dioxide, thereby improving the atmospheric environment and slowing down the generation of acid rain;  Secondly, there are the adverse impacts caused by the terminal processing of the LAP, as mentioned earlier; And thirdly, in the face of the increasing CO2 emissions, the whole society will take appropriate countermeasures, for example, by optimizing energy use, improving production processes, introducing more environmentally friendly technologies, etc. Introducing and implementing a more stringent environmental protection system, including controlling carbon emissions from industries and the energy sector, promoting the development of renewable energy sources, fostering green scientific and technological innovations, financial incentives for the low-carbon economy, etc. To further reduce CO2 emissions.

Industrial LAP emissions: The industrial sulfur dioxide removal rate and industrial fume and dust removal rate of LAP emissions in PRD urban agglomeration showed a three-stage multi-step downward trend, with multiple fluctuation steps in the middle stage of development, and then to the third stage of regional stability, indicating that the industrial sulfur dioxide removal rate and the industrial fume and dust removal rate affect the LAP emission reduction. The industrial sulfur dioxide removal rate of the YRD urban agglomeration LAP emissions showed an "inverted U" type, indicating that the industrial sulfur dioxide removal rate did not affect LAP emission reduction. The industrial fume and dust removal rate. In the YRD urban agglomeration, LAP emissions showed a three-stage multi-step downward trend, with multiple fluctuation steps in the intermediate stage of development and then to the third stage of regional stability. This indicates that the industrial fume and dust removal rate affects LAP emission reduction.

3.3.5. Trade effect factors-Total trade imports and exports (TR1)

Industrial CO2 emissions: Total trade imports and exports show a three-step upward trend in CO2 emissions in both PRD and YRD urban agglomeration, from a slow initial effect to an intermediate stage that shows a steep rise and then to a steady state in the third stage. Total trade imports and exports, i.e., the sum of total trade imports and exports, this value can be used to observe the total scale of a country in terms of foreign trade. The above illustrates that as the total scale of foreign trade increases, CO2 emissions increase for two reasons: firstly, the increase in the total scale of foreign trade itself will promote the production activities of enterprises, which will lead to an increase in energy consumption and hence CO2 emissions; Secondly, some foreign enterprises will transfer pollution-intensive industries to the PRD and YRD urban agglomeration through pollution transfer, which will lead to an increase in energy consumption, and hence an increase in CO2 emissions[5].

Industrial LAP emissions: Total trade imports and exports show a three-stage downward trend in LAP emissions in both the PRD and YRD urban agglomeration, from a slow initial effect to a steep decline in the middle stage to a steady state in the third stage. The reason for the decrease in LAP emissions is that, with the increase in the total scale of foreign trade and the introduction of capital and technology from foreign-funded enterprises, the industrial sector is able to improve its production equipment to meet the requirements of "low energy consumption" and "high efficiency". Some foreign-funded enterprises also choose to trade with enterprises that comply with local environmental technologies or regulations, thus promoting more enterprises to upgrade their pollutant reduction [28].

Point 6: In Table 4, the number of digits after the decimal point should be consistent.

Response 6: Thanks a lot for your comments. Your suggestions have been very helpful. We've made the numbers in the charts to the fourth decimal place, and we've made the accuracy of the numbers consistent throughout the text. The corrections are as follows:

Lines 290-291:

Table 3. Importance of factors influencing industrial CO2 and LAP emissions

Region

Scale effect factors

Structural effect factors

Technology effect factors

Governance effect factors

Trade effect factors

PRD

Indicators

SC1

ST1

TE1

TE2

GO1

GO2

GO3

TR1

LAP

0.1652

0.1216

0.1681

0.0256

0.0227

0.1891

0.1427

0.1650

CO2

0.1780

0.0464

0.1915

0.1104

0.0328

0.1578

0.1498

0.1333

YRD

Indicators

SC1

ST1

TE1

TE2

GO1

GO2

GO3

TR1

LAP

0.1724

0.1481

0.1183

0.1862

0.0153

0.0906

0.1712

0.0979

CO2

0.1398

0.0533

0.1507

0.1415

0.1245

0.1169

0.1151

0.1581

Note: is the common main effect factor for CO2 and LAP emissions, is the main effect factor for LAP emissions, is the main effect factor for CO2 emissions

Point 7: There is too much content in Section 4.2. Policy Implications and should be simplified.

Response 7: Thank you for the valuable suggestions. Your suggestions are very inspiring to us. We not only have simplified the article's countermeasures and suggestions in a reasonable and effective way but also maintain the usability of them. The simplifications are as follows:

Lines 560-589:

4.2. Policy Implications

(1) By analyzing the time-dynamic evolution of industrial CO2 and LAP emissions and intensities, it can be seen that the PRD and YRD urban agglomeration are both more effective in managing industrial LAP. However, management of industrial CO2 still needs to be further strengthened, so both the PRD and YRD urban agglomeration should focus more on the "Carbon Reduction" perspective in their efforts to reduce carbon emissions. Therefore, both the PRD and YRD should focus more on "carbon reduction" and give full play to their synergistic effects in industrial CO2 reduction to help achieve the goal of "dual carbon".

(2) Based on the analysis of the importance of factors influencing industrial CO2 and LAP emissions, both PRD urban agglomeration and YRD urban agglomeration should pay attention to the influence of the scale effect. Firstly, the industrial sector should scientifically and efficiently analyze the benefits of economies of scale; Secondly, in the past, many industrial sectors have adopted a sloppy mode of economic growth, which has led to the high consumption of resources, slow capital turnover, high losses and wastage, and low economic efficiency, and so on. We should strive to achieve intensive economic growth.

For the PRD urban agglomeration, the industrial sector should pay attention to dust and sulphur removal and promote the innovation of equipment and technology. Government departments can promote the impact of foreign trade on pollution and carbon reduction by formulating environmental protection standards, engaging in international co-operation, incentivizing the adoption of cleaner technologies, and strengthening regulation to achieve a balance between sustainable economic development and environmental protection.

For the YRD urban agglomeration, the industrial departments should pay attention to the upgrading and transformation of the industrial structure, optimize the allocation of resources among industries, vigorously promote the strategy of scientific and technological innovation, reduce the number of highly polluting industries, increase the number of high-tech industries and reduce energy consumption. Suggestions for the work of government departments are similar to those for the PRD urban agglomeration.

Point 8: There are some syntax errors, sentence errors and other English Language errors that must be corrected, e.g. “for example, Tang Xiangbo et al. [6] For example, Tang Xiangbo et al.”; Line 111, “In this thesis”. Possibly, a native English Language speaking Scientists should be employed for the final editing.

Response 8: Thanks a lot for your comments and recommendations. We are very sorry for some grammatical and expression errors. After careful verification, we have corrected them one by one, and will check the article more carefully to enhance the rigor of the article. The specific corrections are as follows:

Lines 75-78: For example, Tang et al.[5] studied the spatial evolution law and mechanism of the influencing factors of the synergistic effect of pollution and carbon reduction based on China's provincial-level data through a spatiotemporal geographically weighted regression model.

Point 9: The format of references should be consistent, e.g. Line 443, 464, 490, 516, 522, 533, 544, the page number is missed. Line 548, “Huang Ruxia, Zhong Qiumeng, Wu Xiaohui”; Line 558, “Li Z.J., Xu J.L., Wang J”; Line 596, “FRONTIERS OF ENVIRONMENTAL SCIENCE & ENGINEERING”; etc.

Response 9: Thank you very much for your suggestion. Corrections have been made in response to your kind suggestion. The formatting of all references has been standardized and made consistent. The specific corrections are as follows:

Lines 606-668:

  1. Chen X, Z.J., Tang X. Synergistic effect of industrial pollution and carbon reduction in China and its influence mechanism. Resource Science 2022, 44, 2387-2398, doi:10.18402/resci.2022.12.01.
  2. Huang R, Z.Q., Wu X et al. Impacts of Economic Structural Transition on Synergetic Control of CO2 and Air Pollutants in Guangdong Province. Research of Environmental Sciences 2022, 35, 2303-2311, doi:10.13198/j.issn.1001-6929.2022.02.30.
  3. Xiang M, W.S., Lv L, et al. Synergistic Paths of Reduced Pollution and Carbon Emissions Based on Different Power Demands in China. Environmental Science 2023, 44, 3637-3648, doi:10.13227/j.hjkx.202207256.
  4. Jiang Y, T.X., Ren K et al. . Research on the drivers of pollution and carbon reduction in China based on double-layer nested SDA. Systems Engineering-Theory & Practice 2022, 44, 3294-3304, doi:10.12011/SETP2022-0408.
  5. Tang X, Z.Y., Cao L, et al. Analysis of the spatial and temporal characteristics of the synergistic effect of pollution and carbon reduction in China and its influence mechanism. Research of Environmental Sciences 2022, 35, 2252-2263, doi:10.13198/j.issn.1001-6929.2022.08.10.
  6. Liu M, L.S., Li J et al. Assessment and prediction of synergistic effect of pollution and carbon reduction in Tianjin. China Environmental 2022, 42, 3940-3949, doi:10.19674/j.cnki.issn1000-6923.20220329.004.
  7. Li Z, X.J., Wang J, et al. Spatial and Temporal Heterogeneity of Urban Carbon Emissions and Their Influencing Factors in Yangtze River Economic Belt. Resources and Environment in the Yangtze Basin 2023, 32, 525-536, doi:10.11870/cjlyzyyhj202303008.
  8. Yuan W, S.H., Wang J,et al. Spatial-temporal Evolution and Driving Forces of Urban Pollution and Carbon Reduction in China. ECONOMIC GEOGRAPHY 2022, 42, doi:10.15957/j.cnki.jjdl.2022.10.009.
  9. Liu W, M.X., Li W et al. Study on decoupling relationship between industrial growth and carbon dioxide emission in the urban agglomeration in the Yellow River Basin. Journal of Environmental Engineering Technology 2023, 13, 849-856, doi:10.12153/j.issn.1674-991X.20220461.
  10. Kang Z, L.W., Liu W. Influencing factors and promoting measures of industrial pollution abatement and carbon reduction of the city clusters in the Yellow River basin. China Environmental Science 2023, 43, 1946-1956, doi:10.19674/j.cnki.issn1000-6923.2023.0046.
  11. Wang F, K.D., Zhu X. Spatial-Temporal Evolution and Driving Force Analysis of Industrial Air Pollutants and Carbon Dioxide Emission Reduction from the Yangtze River Delta. Research of Environmental Sciences 2024, 37, 1-17, doi:10.13198/j.issn.1001-6929.2024.01.08.
  12. Zhang W, X.Y., Hui J. The spatio-temporal impacts and driving factors of the synergistic effects of reducing pollution and carbon emissions in cities of China. Chinese Journal of Environmental Management 2023, 15, 38-47, doi:10.16868/j.cnki.1674-6252.2023.02.038.
  13. Song D, C., Wang B. How Environmental Equity Trading Achieves Pollution Reduction and Carbon Synergies: Theory and Empirical Evidence. Journal of Quantitative & Technological Economics 2022, 1-21, doi:10.13653/j.cnki.jqte.20231214.004.
  14. Guo Q, L.J. Import trade, technology spillover and China’s carbon emissions. China Population, Resources and Environment 2013, 23, 105-109, doi:10.3969/j.issn.1002-2104.2013.03.017.
  15. Fujii H, M.S., Kaneko S. Decomposition analysis of air pollution abatement in China: empirical study for ten industrial sectors from 1998 to 2009. Journal of Cleaner Production 2013, 59, 22-31, doi:10.1016/j.jclepro.2013.06.059.
  16. Lyu W, L.Y., Guan D, et al. . Driving forces of Chinese primary air pollution emissions: an index decomposition analysis. Journal of Cleaner Production 2016, 133, 136-144, doi:10.1016/j.jclepro.2016.04.093.
  17. Li M, W.Q. Study on the Technology Progress Impact on Pollutant Generation Based on Malmquist Index: Take Industry SO2 as an Example. Acta Scientiarum Naturalium Unversitatis Pekinensis 2012, 48, 817-823, doi:10.13209/j.0479-8023.2012.106.
  18. Liu Q, W.Q., Liu Y. Study on the relationship between economic growth, international trade and pollution emissions: an empirical analysis based on SO2 emissions in the United States and China. China Population, Resources and Environment 2012, 22, 170-176, doi:10.3969/j.issn.1002-2104.2012.05.028.
  19. P, Z. End-of-pipe or process-integrated: evidence from LMDI decomposition of China's SO2 emission density reduction. FRONTIERS OF ENVIRONMENTAL SCIENCE & ENGINEERING 2013, 7, 867-874, doi:10.1007/s11783-013-0541-0.
  20. Cui D, J.B. Comprehensive evaluation of water ecological civilization based on random forests regression algorithm. Advances in Science and Technology of Water Resources 2014, 56-60,79, doi:10.3880 /j.issn.1006-7647.2014.05.011.
  21. Zhao Long, S.G., Wu Wei et al. Rainfall Prediction Model for Mountain Torrent Disaster Based on Random Forest Regression Algorithm. Journal of University of Jinan (Science and Technology) 2022, 36, 404-411,423, doi:10.13349/j.cnki.jdxbn.20220325.001.
  22. Cheng Y, X.C., Ren J et al. Atmospheric Environment Effect of Industrial Structure Evolution in Shandong Province. CHINA POPULATION,RESOURCES AND ENVIRONMENT 2014, 24, 157-162, doi:10.3969/j.issn.1002-2104.2014.01.022.
  23. Bian Y, L.X., Zhou X,et al. Temporal and spatial evolution characteristics and influencing factors of industrial carbon emissions in Beijing-Tianjin-Hebei region. Environmental Science & Technology 2021, 44, 37-47, doi:10.19672/j.cnki.1003-6504.1329.21.338.
  24. Wang Z, W.L. A Study on the Relationship Among R&D Input, Upgrading Industrial Structure and Carbon Emission. Journal of Industrial Technological Economics 2019, 38, 62-70, doi:10.3969/j.issn.1004-910X.2019.05.008.
  25. Yu W, Z.T., Shen D. County-levelspatial pattern and influencing factors evolution of carbon emission intensity in China: A random forest model analysis. China Environmental Science 2022, 42, 2788-2798, doi:10.19674/j.cnki.issn1000-6923.20220219.001.
  26. Xu K, W.Y. The Influence of China’s OFDI on Its Domestic CO2 Emissions: An Empirical Analysis Based on China’s Provincial Panel Data. GUOJI SHANGWU YANJIU 2015, 76-86, doi:10.13680/j.cnki.ibr.2015.01.008.
  27. Duan J, Y.J. Wet desulphurisation and carbon emission. Thermal Power Generation 2011, 40, 83-84.
  28. Chen Y, L.K. Factor decomposition of SO2 emission intensity in Chinese industrial sectors and its influencing factors--an analysis based on the backward and forward linkage of FDI industries. Management World 2010, 14-21, doi:10.19744/j.cnki.11-1235/f.2010.03.003.

Round 2

Reviewer 1 Report

Comments and Suggestions for Authors

This paper has been revised according to the comments and suggestions, but there are still the following problems:

1. The innovation of the paper is relatively weak.

2. Almost all the references in this paper come from Chinese literature. But more English literature is needed, and more international research status should be described and explained.

3. Why is negative for the CO2 and LAP emission intensity in Figure 2?

Author Response

Response to Reviewer 1’s Comments

Article title: “Impact Factors of Industrial Pollution and Carbon Reduction under the "Dual Carbon" Target: A Case Study of the Pearl River Delta and Yangtze River Delta Urban Aggregation

Review report:

This paper has been revised according to the comments and suggestions, but there are still the  

following problems:

  1. The innovation of the paper is relatively weak.
  2. Almost all the references in this paper come from Chinese literature. But more English literature is needed, and more international research status should be described and explained.
  3. Why is negative for the CO2 and LAP emission intensity in Figure 2?

Response: Thank you very much for your kind and constructive comments on our manuscript. Your comments significantly improved the quality of the manuscript. We have revised the manuscript point by point based on your feedback. Please see the detailed response below.

Point 1: The innovation of the paper is relatively weak.

Response 1: Thank you very much for your suggestion. Based on your comments, we have highlighted the innovations of this study, please see line 87-123 in the revised manuscript.

Lines 87-123:

Generally speaking, the following deficiencies remain: (1) Pollution and carbon reduction at the industrial level is one of the key areas of work in the "dual-carbon" program, but existing research has paid little attention to this area, with most of the results focusing on the factors influencing the emission reduction of greenhouse gases or air pollutants, which is insufficient to meet the current development; (2) up to the present time, research on the factors influencing the reduction of pollution and carbon in the Pearl River Delta and Yangtze River Delta urban agglomerations has not been sufficiently explored. YRD urban agglomeration needs to be sufficiently explored, and there needs to be more research on inter-provincial heterogeneity to provide management decision support for local governments to formulate differentiated pollution and carbon reduction policies.

Therefore, we used data on industrial CO2 and LAP from 2002 to 2021 and applied a random forest regression model to systematically study the influencing factors of pollution and carbon reduction in the PRD and YRD urban agglomeration. The main objectives are: (1) to identify the main influencing factors for industrial pollution reduction and carbon reduction in PRD and YRD urban agglomeration; (2) to explore the temporal dynamic evolution patterns of industrial CO2 and LAP emissions and intensities in PRD and YRD urban agglomeration; (3) to provide a more comprehensive and precise reference for the formulation of environmental protection policies for the urban agglomerations of PRD and YRD urban agglomeration. This approach contributes to an in-depth understanding of the interactions of various factors in pollution and carbon reduction and provides a scientific basis for future sustainable urban development.

This paper presents three new contributions: (1) it discusses pollution reduction and carbon reduction in the same framework; (2) it is significant for the PRD and YRD urban agglomeration to fully utilize their synergistic effects in industrial pollution reduction and carbon reduction. The PRD and YRD urban agglomeration play a key role in China's economic and social development and simultaneously face unique environmental challenges. This choice considers various factors such as concentration of economic activities, different energy mixes, geographical characteristics, and differences in potential policy implementation. The two regions are similar regarding high economic development and urbanization but differ in geographic location and industrial composition. In addition, recognizing the potential inter-provincial heterogeneity of the two clusters, including differences in environmental policies, resource distribution, and economic status, this thesis aims to provide information on the similarities and differences between the selected clusters in terms of industrial CO2 and LAP emission reductions. (3) It provides a methodology to investigate how to achieve synergistic development of pollution reduction and carbon reduction in multiple aspects, including energy, industrial structure, production technology, science, technology and innovation, and trade.

Point 2: Almost all the references in this paper come from Chinese literature. But more English

literature is needed, and more international research status should be described and explained.

Response 2: Thanks for your comments and recommendation. Based on your comments, we consulted more English literature, explained and integrated previous research in more depth. For the shortcomings of previous studies, we improved them in our article, highlighted our innovative points, please see line 65-86 in the revised manuscript.

Lines 65-86:

By consulting the relevant literature, scholars have researched the influencing factors of pollution and carbon reduction from different angles to achieve the pollution and carbon reduction goal. The perspective of the research level can be summarized at the national level [1-4], the local level [5,6], and the industry sector level [7,8]. The research method perspective can mainly be attributed to the decomposition analysis method, the analysis method including the index decomposition analysis method (IDA) and structural decomposition analysis method (SDA)[9,10]. For example, Yan et al. used the IDA method to identify the main influencing factors of the overall change in CO2 emissions from provincial thermal power generation in China[11]. Wang et al. used the SDA method to explore the driving factors of carbon emissions changes in China's domestic trade from the regional scale[12]. In addition, some scholars also used econometric models to study the influencing factors of pollution and carbon reduction[13]. For example, Xian et al. Based on the data of China, through the double difference model (DID), this paper evaluates the synergistic effect and mechanism of carbon trading pilot policy on carbon and air pollutant emissions in the whole country and in the power, industry, transportation, and residential sectors[14]. Zhu et al. analyzed the influencing factors of hidden carbon emissions in China's construction industry based on the extended STIRPAT model [15]. Lin et al. applied the Spatial Durbin Error Model (SDEM) to study the socioeconomic factors that can synergistically affect CO2 and PM2.5 emissions [16]. In selecting indicators, scholars mainly attributed the influencing factors of pollution and carbon reduction to the following aspects: scale effect, structural effect, technical effect, governance factors, trade factors and population factors[17-19].

Point 3: Why is negative for the CO2 and LAP emission intensity in Figure 2?

Response 3: Thank you for your comment. According to your comments, we carefully checked figure 2 and found that there was no error in the original picture. According to the legend of the chart and the right coordinate, it can be seen that all the CO2 and LAP emission intensities are greater than 0, without any unreasonable negative numbers. But you gave us a good suggestion, so we changed the expression of the chart. please see line 262-263 in the revised manuscript.

Lines 262-263:

Figure 2before Analysis of the time-dynamic evolution of industrial CO2 and LAP emissions

Figure 1after Analysis of the time-dynamic evolution of industrial CO2 and LAP emissions and intensities.

Reviewer 3 Report

Comments and Suggestions for Authors

The author needs to double-check on page 4 line 170 with "Error! Reference source not found".

Author Response

Response to Reviewer 3’s Comments

Article title: “Impact Factors of Industrial Pollution and Carbon Reduction under the "Dual Carbon" Target: A Case Study of the Pearl River Delta and Yangtze River Delta Urban Aggregation

Review report:

The author needs to double-check on page 4 line 170 with "Error! Reference source not found".

Response: Thank you very much for your kind and constructive comments on our manuscript. Your comments significantly improved the quality of the manuscript. We have revised the manuscript point by point based on your feedback. Please see the detailed response below.

Main Comments:

Point 1: The author needs to double-check on page 4 line 170 with "Error! Reference source not found".

Response 1: Thanks a lot for your comments and recommendations. Sorry for the formatting error in the reference. We have updated and corrected the references in the literature. Here are our updates:

Lines 166-168:

Industrial production activities are an important pathway for industrial CO2 and LAP emissions, so there is an inseparable relationship between their scale and carbon reduction[22].

Reviewer 4 Report

Comments and Suggestions for Authors

The comment on references has not been answered carefully. From the response 2, the specific modifications are not clear, the  references which are deleted, newly added, etc. Please refer to the previous second comment. 

Comments on the Quality of English Language

Minor editing of English language required.

Author Response

Response to Reviewer 4’s Comments

Article title: “Impact Factors of Industrial Pollution and Carbon Reduction under the "Dual Carbon" Target: A Case Study of the Pearl River Delta and Yangtze River Delta Urban Aggregation

Review report:

The comment on references has not been answered carefully. From the response 2, the specific modifications are not clear, the references which are deleted, newly added, etc. Please refer to the previous second comment.

Response: Thank you very much for your kind and constructive comments on our manuscript. Your comments significantly improved the quality of the manuscript. We have revised the manuscript point by point based on your feedback. Please see the detailed response below.

Main Comments:

Point 1: The comment on references has not been answered carefully. From the response 2, the specific modifications are not clear, the references which are deleted, newly added, etc. Please refer to the previous second comment.

Response 1: Thanks a lot for your comments and recommendations. You mentioned that ‘The references on pollution reduction and carbon reduction are insufficient, some references should be cited and analyzed’, We have cited more references based on your kind suggestion and made more analysis based on previous references. We have added more papers about pollution reduction and carbon reduction, and made a more detailed explanation for this. Specific corrections are as follows:

Lines 65-86:

By consulting the relevant literature, scholars have researched the influencing factors of pollution and carbon reduction from different angles to achieve the pollution and carbon reduction goal. The perspective of the research level can be summarized at the national level [1-4], the local level [5,6], and the industry sector level [7,8]. The research method perspective can mainly be attributed to the decomposition analysis method, the analysis method including the index decomposition analysis method (IDA) and structural decomposition analysis method (SDA)[9,10]. For example, Yan et al. used the IDA method to identify the main influencing factors of the overall change in CO2 emissions from provincial thermal power generation in China[11]. Wang et al. used the SDA method to explore the driving factors of carbon emissions changes in China's domestic trade from the regional scale[12]. In addition, some scholars also used econometric models to study the influencing factors of pollution and carbon reduction[13]. For example, Xian et al. Based on the data of China, through the double difference model (DID), this paper evaluates the synergistic effect and mechanism of carbon trading pilot policy on carbon and air pollutant emissions in the whole country and in the power, industry, transportation, and residential sectors[14]. Zhu et al. analyzed the influencing factors of hidden carbon emissions in China's construction industry based on the extended STIRPAT model [15]. Lin et al. applied the Spatial Durbin Error Model (SDEM) to study the socioeconomic factors that can synergistically affect CO2 and PM2.5 emissions [16]. In selecting indicators, scholars mainly attributed the influencing factors of pollution and carbon reduction to the following aspects: scale effect, structural effect, technical effect, governance factors, trade factors and population factors[17-19].

Round 3

Reviewer 1 Report

Comments and Suggestions for Authors

The overall quality of this paper has been greatly improved through the revision. The structure of the paper is more logical, and the description of the innovative points is clearer. Please read through and check this paper further.

Author Response

Response to Reviewer 1’s Comments

Article title: “Impact Factors of Industrial Pollution and Carbon Reduction under the "Dual Carbon" Target: A Case Study of the Pearl River Delta and Yangtze River Delta Urban Aggregation

Review report:

The overall quality of this paper has been greatly improved through the revision. The structure of the paper is more logical, and the description of the innovative points is clearer. Please read through and check this paper further.

Response: Thank you very much for your kind and constructive comments on our manuscript. Your comments significantly improved the quality of the manuscript. We have revised the manuscript point by point based on your feedback. Please see the detailed response below.

Point 1: The overall quality of this paper has been greatly improved through the revision. The structure of the paper is more logical, and the description of the innovative points is clearer. Please read through and check this paper further.

Response 1: Thank you very much for your feedback, we are very happy to know that you are satisfied with the overall improvement of the article. We will read it carefully and check it further. Here are a few important corrections:

original manuscript:

Lines 83-98:

On the whole, although the existing research has laid the foundation for reducing pollution and carbon, and has achieved some results, there are still some shortcomings. First of all, the reduction of pollution and carbon at the industrial level is a key point of the "double carbon" work, but the existing studies still pay little attention to this aspect, and most of the results only focus on the influence factors of greenhouse gas or air pollutant reduction, which cannot meet the existing development. Moreover, up to now, the research on the influence factors of pollution reduction and carbon reduction in the PRD and YRD urban agglomerations has not been sufficiently explored, and the research on inter-provincial heterogeneity is insufficient to provide management decision support for local governments to formulate differentiated policies on pollution reduction and carbon reduction.

Based on this, this thesis adopts the random forest regression model method, selects the PRD and YRD urban agglomerations as the study area, and identifies the main influencing factors of industrial pollution reduction and carbon reduction by analyzing the data of industrial CO2 and industrial LAP from 2000 to 2020 and gives related countermeasure suggestions.

revised manuscript:

Lines 86-123:

In general, the following shortcomings persist: (1) The reduction of pollution and carbon emissions at the industrial level is a crucial aspect of the 'dual-carbon' program. However, existing research has largely neglected this area, with most studies focusing on the factors that influence the reduction of greenhouse gases or air pollutants. This is insufficient to meet current development needs. (2) To date, research on the factors that influence the reduction of pollution and carbon in the urban agglomerations of the Pearl River Delta and Yangtze River Delta has been inadequate. Further exploration of the YRD urban agglomeration is necessary, along with additional research on inter-provincial heterogeneity. This will provide local governments with the necessary information to formulate differentiated pollution and carbon reduction policies.

Therefore, we analyzed data on industrial CO2 and LAP from 2002 to 2021 and used a random forest regression model to systematically study the factors influencing pollution and carbon reduction in the PRD and YRD urban agglomerations. The main objectives are: (1) The aim of this study is to identify the main factors that influence industrial pollution reduction and carbon reduction in the PRD and YRD urban agglomerations. (2) Additionally, we aim to explore the temporal dynamic evolution patterns of industrial CO2 and LAP emissions and intensities in these areas. (3) The results of this study will provide a more comprehensive and precise reference for the formulation of environmental protection policies for the PRD and YRD urban agglomerations. This approach contributes to a comprehensive understanding of the interactions between various factors in pollution and carbon reduction. It provides a scientific basis for future sustainable urban development.

This paper presents three new contributions. Firstly, it discusses pollution reduction and carbon reduction in the same framework. Secondly, it highlights the significance of fully utilizing the synergistic effects of the PRD and YRD urban agglomerations in industrial pollution reduction and carbon reduction. Finally, it emphasizes that the PRD and YRD urban agglomerations play a key role in China's economic and social development while simultaneously facing unique environmental challenges. This choice takes into account several factors, including the concentration of economic activities, differences in energy mixes, geographical characteristics, and potential policy implementation. The two regions share similarities in terms of high economic development and urbanization, but differ in their geographic location and industrial composition. Additionally, this thesis aims to provide information on the similarities and differences between the selected clusters in terms of industrial CO2 and LAP emission reductions, while acknowledging the potential inter-provincial heterogeneity of the two clusters, including differences in environmental policies, resource distribution, and economic status. Thirdly, it provides a methodology to investigate how to achieve synergistic development of pollution reduction and carbon reduction in multiple aspects, including energy, industrial structure, production technology, science, technology and innovation, and trade.

Reviewer 4 Report

Comments and Suggestions for Authors

The manuscript should be revised carefully according to the first review. 

Comments on the Quality of English Language

Moderate editing of English language required.

Author Response

Response to Reviewer 4’s Comments

Article title: “Impact Factors of Industrial Pollution and Carbon Reduction under the "Dual Carbon" Target :A Case Study of the Pearl River Delta and Yangtze River Delta Urban Aggregation

Review report:

The manuscript should be revised carefully according to the first review.

Review the first report:

Energy and environmental issues have become important factors in the global economic and social development. The study is interesting and meaningful. However, there were some issues that should be improved in this manuscript.

  1. In the Introduction section, the "Double Carbon" policies of other major countries should be briefly introduced.
  2. The references on pollution reduction and carbon reduction are insufficient, some references should be cited and analyzed, e.g., DOI: 10.1016/j.fuel.2023.129925; 10.1016/j.mineng.2023.108296.
  3. There are some errors, e.g. Lines 328, 354, 356, 394, 415, “Error! Reference source not found.”; Line 64, “national level[1] local level”; etc.
  4. The order of figures are wrong. It is suggested that Figures 1-4 (Pages 9-12) should be merged.
  5. The titles of Sections 3.3.1-3.3.5 should be revised.
  6. In Table 4, the number of digits after the decimal point should be consistent.
  7. There is too much content in Section 4.2. Policy Implications and should be simplified.
  8. There are some syntax errors, sentence errors and other English Language errors that must be corrected, e.g. “for example, Tang Xiangbo et al.[6] For example, Tang Xiangbo et al.”; Line 111, “In this thesis”. Possibly, a native English Language speaking Scientists should be employed for the final editing.
  9. The format of references should be consistent, e.g. Line 443, 464, 490, 516, 522, 533, 544, the page number is missed. Line 548, “Huang Ruxia, Zhong Qiumeng, Wu Xiaohui”; Line 558, “Li Z.J., Xu J.L., Wang J”; Line 596, “FRONTIERS OF ENVIRONMENTAL SCIENCE & ENGINEERING”; etc.
  10. Moderate editing of English language required.

Response: Thank you very much for your comments. We have carefully considered your suggestions and have made careful revisions to ensure that the quality of the manuscript is improved. According to your suggestion, we have corrected the grammatical and editorial mistakes throughout the manuscript. Moreover, we have invited an English native speaker to improve the English language of the revised manuscript.  We have focused on the issues mentioned in the first review and have revised the paper in detail. Please see the detailed response below.

Main Comments:

Point 1: In the Introduction section, the "Double Carbon" policies of other major countries should be briefly introduced.

Response 1: Thank you very much for your suggestions. Based on your comments, we have revised the introduction to include a brief description of the "dual carbon" policies of major countries such as the US, EU and India. This will provide more comprehensive background information and enrich the content of the paper. Here is the section we have modified:

Lines 32-64:

Energy and environmental issues are crucial factors in global economic and social development. There is an international consensus to address climate change and develop a low-carbon economy proactively. China, being the largest country in energy consumption and carbon emissions, has a significant responsibility to reduce pollution and carbon emissions. On 22 September 2020, Xi Jinping, General Secretary of the Communist Party of China Central Committee, Chinese President and Chairman of the Central Military Commission, delivered a speech at the general debate of the 75th United Nations General Assembly. He announced that China aims to peak carbon dioxide emissions by 2030 and achieve carbon neutrality by 2060. This marks China's inaugural commitment to the United Nations stage, a significant development in global climate governance. The 'dual-carbon' goal was subsequently incorporated into China's 14th Five-Year Plan and 2035 Vision Outline. On February 26th, 2021, the National Development and Reform Commission (NDRC), the State-owned Assets Supervision and Administration Commission of the State Council (SASAC), and others issued the 'Guiding Opinions on Strengthening Green and Low-Carbon Transformation and Development.' The goal proposed is to achieve peak carbon dioxide emissions before 2030. The 'dual-carbon' goal aims to promote China's high-quality development and the construction of ecological civilization. This concept emphasizes the harmonious coexistence of human beings with nature, sustainable development, and environmental protection. It is a comprehensive development concept in China. Additionally, countries around the world are implementing various 'dual-carbon' policies to address climate change and achieve carbon peak and neutrality. For instance, in the United States, these policies involve rejoining the Paris Agreement, setting ambitious carbon-neutral targets for 2050, and making significant investments in clean energy and infrastructure. The administration's policy emphasizes comprehensive climate action, including concerns about specific sector objectives, policy reversals, and cooperation with domestic and international partners [1]. The European Union's 'Green Deal' is an ambitious plan to transform its member states into carbon-neutral economies by 2050. This plan addresses the energy transition, promotion of renewable energy, and industrial innovation [2]. Despite its energy-intensive industries, India is aggressively pursuing carbon reduction targets and investing in renewable energy and clean technologies for sustainable development [3]. These policies reflect the global consensus on cooperation in tackling climate change and highlight the challenges faced by countries in achieving carbon neutrality.

Point 2: The references on pollution reduction and carbon reduction are insufficient, some references should be cited and analyzed, e.g., DOI: 10.1016/j.fuel.2023.129925; 10.1016/j.mineng.2023.108296.

Response 2: Thank you very much for your suggestion. We have added more references based on your comments, which include some international research results such as DOI:10.1016/j.jenvman.2023.118525、10.3390/ijerph182413307和10.1007/s10584-011-0034-8 and so on. These additional references will provide fuller support and context for our paper and help to analyze the issues related to pollution abatement and carbon reduction in greater depth. Specific corrections are as follows:

Lines 65-85:

Scholars have researched the factors influencing pollution and carbon reduction from various perspectives to achieve the goal of reducing pollution and carbon emissions. The research has been conducted at the national level [4-7], the local level [8,9], and the industry sector level [10,11]. The research method perspective is mainly attributed to the decomposition analysis method, which includes the index decomposition analysis method (IDA) and the structural decomposition analysis method (SDA) [12,13]. For instance, Yan et al. (2019) used the IDA method to identify the main influencing factors of the overall change in CO2 emissions from provincial thermal power generation in China [14]. Wang et al. (2019) employed the SDA method to investigate the driving factors behind changes in carbon emissions resulting from domestic trade in China at a regional scale [15]. Other scholars have also utilized econometric models to examine the factors that influence pollution and carbon reduction [16]. For instance, Xian et al. (2019) evaluated the synergistic effect and mechanism of the carbon trading pilot policy on carbon and air pollutant emissions in China using the double difference model (DID). The study covered the power, industry, transportation, and residential sectors [17]. Zhu et al. (2020) analyzed the factors influencing hidden carbon emissions in China's construction industry using the extended STIRPAT mode [18]. Lin et al. (year not provided) used the Spatial Durbin Error Model (SDEM) to investigate the socioeconomic factors that can jointly impact CO2 and PM2.5 emissions [19]. Scholars have identified the following aspects as the main influencing factors of pollution and carbon reduction: scale effect, structural effect, technical effect, governance factors, trade factors, and population factors [20-22].

Lines 604-648:

  1. Chen, X.H.; Tee, K.; Elnahass, M.; Ahmed, R. Assessing the environmental impacts of renewable energy sources: A case study on air pollution and carbon emissions in China. Journal of Environmental Management 2023, 345, 118525, doi:10.1016/j.jenvman.2023.118525.
  2. Zhang, Z.H.; Zhang, J.X.; Feng, Y.C. Assessment of the Carbon Emission Reduction Effect of the Air Pollution Prevention and Control Action Plan in China. International Journal of Environmental Research and Public Health 2021, 18, 13307, doi:10.3390/ijerph182413307.
  3. Akter, S.; Bennett, J. Household perceptions of climate change and preferences for mitigation action: the case of the Carbon Pollution Reduction Scheme in Australia. Climatic Change 2011, 109, 417-436, doi:10.1007/s10584-011-0034-8.
  4. Wang, S.J.; Zhang, S.H.; Cheng, L. Investigating the Synergy between CO2 and PM2.5 Emissions Reduction: A Case Study of China's 329 Cities. Atmosphere 2023, 14, doi:10.3390/atmos14091338.
  5. Chen, S.Y.; Tan, Z.X.; Mu, S.Y.; Wang, J.Y.; Chen, Y.Y.; He, X.W. Synergy level of pollution and carbon reduction in the Yangtze River Economic Belt: Spatial-temporal evolution characteristics and driving factors. Sustainable Cities and Society 2023, 98, 104859, doi:10.1016/j.scs.2023.104859.
  6. Hu, C.; Ma, X.; Yang, L.; Chang, X.; Li, Q. Spatial-temporal variation and driving forces of the synergy of “pollution reduction, carbon reduction, green expansion and economic growth”: evidence from 243 cities in China. 2023, 11, 1-28, doi:10.3389/fevo.2023.1202898.
  7. Zhao, J.Y.; Xi, X.; Na, Q.; Wang, S.S.; Kadry, S.N.; Kumar, P.M. The technological innovation of hybrid and plug-in electric vehicles for environment carbon pollution control. Environmental Impact Assessment Review 2021, 86, 106506, doi:10.1016/j.eiar.2020.106506.
  8. Jiang, Y.; Zhang, Y.; Brenya, R.; Wang, K. How environmental decentralization affects the synergy of pollution and carbon reduction: Evidence based on pig breeding in China. Heliyon 2023, 9, e21993, doi:10.1016/j.heliyon.2023.e21993.
  9. Andreoni, V.; Galmarini, S. Drivers in CO2 emissions variation: A decomposition analysis for 33 world countries. Energy 2016, 103, 27-37, doi:10.1016/j.energy.2016.02.096.
  10. Wang, S.J.; Zhang, S.H.; Cheng, L. Drivers and Decoupling Effects of PM2.5 Emissions in China: An Application of the Generalized Divisia Index. International Journal of Environmental Research and Public Health 2023, 20, doi:10.3390/ijerph20020921.
  11. Yan, Q.Y.; Wang, Y.X.; Balezentis, T.; Sun, Y.K.; Streimikiene, D. Energy-Related CO2 Emission in China's Provincial Thermal Electricity Generation: Driving Factors and Possibilities for Abatement. Energies 2018, 11, 1096, doi:10.3390/en11051096.
  12. Wang, Z.H.; Li, Y.M.; Cai, H.L.; Yang, Y.T.; Wang, B. Regional difference and drivers in China's carbon emissions embodied in internal trade. Energy Economics 2019, 83, 217-228, doi:10.1016/j.eneco.2019.06.023.
  13. Wang, Z.; Shao, H. Spatiotemporal differences in and influencing factors of urban carbon emission efficiency in China’s Yangtze River Economic Belt. Environmental Science Pollution Research 2023, 30, 121713-121733, doi:10.1007/s11356-023-30674-7.
  14. Xian, B.; Wang, Y.; Xu, Y.; Wang, J.; Li, X. Assessment of the co-benefits of China's carbon trading policy on carbon emissions reduction and air pollution control in multiple sectors. Economic Analysis and Policy 2024, doi:10.1016/j.eap.2024.01.011.
  15. Zhu, C.; Chang, Y.; Li, X.D.; Shan, M.Y. Factors influencing embodied carbon emissions of China's building sector: An analysis based on extended STIRPAT modeling. Energy and Buildings 2022, 255, 14, doi:10.1016/j.enbuild.2021.111607.
  16. Lin, H.X.; Jiang, P. Analyzing the phased changes of socioeconomic drivers to carbon dioxide and particulate matter emissions in the Yangtze River Delta. Ecological Indicators 2022, 140, 109044, doi:10.1016/j.ecolind.2022.109044.
  17. Zhang, P.D. End-of-pipe or process-integrated: evidence from LMDI decomposition of China's SO2 emission density reduction. Frontiers of Environmental Science & Engineering 2013, 7, 867-874, doi:10.1007/s11783-013-0541-0.
  18. Zha, Q.F.; Liu, Z.; Wang, J. Spatial pattern and driving factors of synergistic governance efficiency in pollution reduction and carbon reduction in Chinese cities. Ecological Indicators 2023, 156, 111198, doi:10.1016/j.ecolind.2023.111198.
  19. Du, Q.; Pang, Q.Y.; Bao, T.N.; Guo, X.Q.; Deng, Y.G. Critical factors influencing carbon emissions of prefabricated building supply chains in China. Journal of Cleaner Production 2021, 280, 12, doi:10.1016/j.jclepro.2020.124398.

Point 3: There are some errors, e.g. Lines 328, 354, 356, 394, 415, “Error! Reference source not found.”; Line 64, “national level[1] local level”; etc.

Response 3: Thank you very much for your detailed correction. We have scrutinized the article for errors in citing sources as well as misrepresentations. We have carefully checked and corrected these errors to ensure that the article meets the high standards required. We will also strengthen our own rigor in future essay writing to prevent the recurrence of such errors. Thank you again for your correction. The revised content is as follows:

original manuscript:

Lines 328 :

this may be due to the adjustment of industrial structure by enterprises, especially to reduce the proportion of pollution-intensive industries, which is an important way to reduce LAP emissions Error! Reference source not found

Lines 354 :

First, there may be a threshold value for the impact of R&D investment on CO2 emissions, below which it will cause an increase in CO 2emissions to rise below a certain threshold. Error! Reference source not found

Lines 394 :

and gradually change the way we manage high "carbon emissions" for low "three wastes" to achieve true "low carbonization". Error! Reference source not found

Lines 415 :

and the chemical reactions in the pollution reduction process can lead to an increase in CO2 emissions emissions. Error! Reference source not found

Lines 64 :

The research level perspectives can be summarized at the national level[1] local level[2]

revised manuscript

Lines 373

This reduction in LAP emissions may be due to enterprises adjusting their industrial structure, particularly by reducing the proportion of pollution-intensive industries. This is an effective method of reducing emissions [26].

Lines 406

Firstly, it is important to note that there may be a threshold for the effect of R&D input on CO2 emissions. If the threshold is lower than a certain level, it can actually cause a rise in CO2 emissions [36];

Lines 451 :

Specifically, it is necessary to adjust the environmental management approach and gradually change the high "carbon emissions" to a low "three wastes" management approach to achieve the real "low carbon" [38].

Lines 478 :

 in the consumption of electricity for the use of the LAP system and in the chemical reactions during the LAP process, all of which can lead to an increase in CO2 emissions [32].

Lines 67 :

The research has been conducted at the national level [4-7], the local level [8,9], and the industry sector level [10,11].

Point 4: The order of figures are wrong. It is suggested that Figures 1-4 (Pages 9-12) should be merged.

Response 4: Thanks a lot for your comments and recommendations. We have rearranged the order of the charts in accordance with your suggestions and combined Figures 1-4 into two charts, i.e. Figures 3-4, based on the Pearl River Delta (PRD) and Yangtze River Delta (YRD) city clusters in order to reduce the length of the charts and to improve their readability. These changes will help make the charts clearer and more structured. Please refer to lines 249-250, lines 262-263, lines 355-358 of the revised draft for details.

Lines 248-249:

Figure 1 Random Forest regression model flowchart

Lines 260-261:

Figure 2 Analysis of the time-dynamic evolution of industrial CO2 and LAP emissions and intensities.

Lines 353-356:

Figure 3. Partial dependence diagram of influencing factors of CO2 and LAP emissions in PRD agglomeration.

Figure 4. Partial dependence diagram of influencing factors of CO2 and LAP emissions in YRD agglomeration.

Point 5: The titles of Sections 3.3.1-3.3.5 should be revised.

Response 5: Thank you for your feedback and recommendations. We have made corrections to the title and we will pay more attention to the formatting, grammar of English writing. Here are the corrections we have made:

Lines 357-526:

3.3.1. Scale effect factors-Industrial value added above the scale (SC1)

Industrial CO2 emissions: The impact of industrial value added above the scale of CO2 emissions in both PRD and YRD urban agglomerations has shown a three-stage upward trend. The trend began with a slow effect, followed by a steep rise in the intermediate stage, and finally reaching a steady state in the third stage. The results of both urban agglomerations illustrate that the effect of industrial value added above the scale on CO2 is divided into two stages: within a certain range, the increase in industrial value added above the scale can lead to increased CO2 emissions. However, after exceeding a certain range, the continual increase in industrial value added above the scale suppresses, to a certain extent, the production activities and the CO2 emissions decline or their growth rate decreases due to the scale effect.

Industrial LAP emissions: The impact of industrial value added above the scale of LAP emissions in both PRD and YRD urban agglomerations showed a three-stage multi-step downward trend, with multiple fluctuating steps in the intermediate stage of development. The impact of industrial value added above the scale in YRD urban agglomeration LAP emissions was weak in the initial stage and declined rapidly after a period of stabilization. This reduction in LAP emissions may be due to enterprises adjusting their industrial structure, particularly by reducing the proportion of pollution-intensive industries. This is an effective method of reducing emissions [26].

3.3.2. Structural effect factors-The proportion of industrial value added in GDP(ST1)

Industrial CO2 emissions: The impact of the proportion of industrial value added in GDP of CO2 emissions in the PRD urban agglomeration shows a "V" shape, indicating that the proportion of industrial value added in GDP does not affect CO2 emission reduction. The influence of the proportion of industrial value added in GDP on CO2 emissions in the YRD urban agglomeration shows a three-stage multi-step downward trend from the initial increase in the proportion of industrial value added in GDP; CO2 emissions also increased. In the middle stage, there was a steep decline in CO2 emissions, followed by a steady state in the third stage. When the proportion of industrial value added to GDP is 48.57%, the maximum impact on CO2 emission reduction is achieved. The above indicates that, with the continuous development of industrialization, due to the continuous innovation of technology and society's increasing concern for environmental sustainability, the highly polluting industries at the beginning will be gradually replaced by less polluting industries, which is also one of the reasons why CO2 emissions in the YRD urban agglomeration appear to increase and then decrease as the proportion of industrial value added in GDP increases.

Industrial LAP emissions: The proportion of industrial value added in GDP shows a three-stage stepped upward trend in LAP emissions in both PRD and YRD urban agglomeration, from the initial slow effect to the intermediate stage showing a steep rise and then to the steady state in the third stage. It indicates that with the continuous development of industry and the increase in energy consumption, LAP emissions will also increase.

3.3.3. Technology effect factors

(1) The proportion of R&D internal expenditure in GDP (TE1)

Industrial CO2 emissions: The proportion of R&D internal expenditure in GDP shows a three-stage upward trend in CO2 emissions in both PRD and YRD urban agglomeration, from a slow effect at the beginning to a steep rise in the middle stage to a steady state in the third stage. "R&D" means research and development. The above results suggest that the proportion of R&D internal expenditure in GDP does not have a significant effect on reducing CO2 emissions, and there may be two reasons for this: Firstly, it is important to note that there may be a threshold for the effect of R&D input on CO2 emissions. If the threshold is lower than a certain level, it can actually cause a rise in CO2 emissions [36]; Secondly, while the investment of scientific research funds has a positive effect on reducing carbon emissions, it may have a lagging effect [37]. It can be observed during the transformation from lower to higher forms that the effect is not always ideal or significant. However, it is important to pursue long-term effects.

Industrial LAP emissions: The proportion of R&D internal expenditure in GDP shows a three-stage multi-step downward trend in LAP emissions in both PRD and YRD urban agglomeration, from the initial slow effect, the existence of multiple fluctuation steps in the intermediate stage of development, and then the regional steady state in the third stage. This is because the effect of LAP emission reduction will become more and more significant with the investment of scientific research funds and the development of green technology.

(2) Energy consumption per 10,000 yuan of industrial value added (TE2)

Industrial CO2 emissions: Energy consumption per 10,000 yuan of industrial value added shows a trend of increasing, then decreasing and then leveling off for PRD urban agglomeration CO2 emissions, and for YRD urban agglomeration CO2 emissions, all show a three-stage stepped downward trend, from the initial slow effect to the intermediate stage showing a steep decline, and then to the third stage of a steady state. The energy consumption per 10,000 yuan of industrial value added refers to the energy consumption per 10,000 yuan of industrial value added, and the above results show that the energy consumption per 10,000 yuan of industrial value added did not affect the CO2 emission reduction.

Industrial LAP emissions: Energy consumption per 10,000 yuan of industrial value added shows a trend of increasing and then leveling off for LAP emissions in PRD urban agglomeration, all show a three-stage stepped upward trend, from the initial slow effect to the intermediate stage showing a steep upward trend, and then to the stable state in the third stage. According to the data analysis, the energy consumption per 10,000 yuan of industrial value added from 2002 to 2021 generally shows a decreasing trend. This indicates that as the energy consumption per 10,000 yuan of industrial value-added decreases, the improvement of energy efficiency has a positive impact on the emission reduction of LAP.

3.3.4. Governance effect factors

(1) The proportion of environmental investment in GDP(GO1)

Industrial CO2 emissions: The impact of the proportion of environmental protection investment on CO2 emissions in the PRD urban agglomeration shows a three-stage, multi-step upward trend, from an initial slow impact, with several fluctuating steps in the intermediate stage of development, to a regional steady state in the third stage; and for the YRD urban agglomeration, a trend of increasing, then decreasing and then leveling off. Some studies show that this is because in dealing with the "three wastes" (i.e. exhaust gas, sewage and industrial waste residues), the relevant departments have adopted an inappropriate treatment method, which leads to a large amount of CO2 emissions when we deal with environmental pollution, resulting in the proportion of environmental protection investment does not have an impact on reducing CO2 emissions. Therefore, we need to make appropriate adjustments to the treatment method. Specifically, it is necessary to adjust the environmental management approach and gradually change the high "carbon emissions" to a low "three wastes" management approach to achieve the real "low carbon" [38].

Industrial LAP emissions: The share of environmental protection investment in LAP emissions in both PRD and YRD urban agglomerations showed a multi-step downward trend, with several fluctuating steps in the intermediate stage of development and then a regional steady state in the third stage. In particular, the YRD urban agglomeration appeared to have a period of stability in the initial stage, indicating that the increase in the proportion of environmental protection investment, the increase in the amount of funds for ecological civilization construction, the continuous development of green technologies and the progress of more advanced pollution treatment facilities had a positive impact on the reduction of LAP emissions.

(2) Industrial sulfur dioxide removal rate (GO2) & Industrial fume and dust removal rate (GO3)

Industrial CO2 emissions: The industrial sulfur dioxide removal rate showed a three-stage upward trend for CO2 emissions in the PRD urban agglomeration, from the initial slow effect to the intermediate stage where there are multiple fluctuating steps and then to the stable state in the third stage. In the YRD urban agglomeration, the removal rate of industrial sulfur dioxide showed a decreasing trend, followed by an increasing trend, and then another decreasing trend. The rate of industrial fume and dust removal showed a three-stage increasing trend for CO2 emissions in the PRD and YRD urban agglomerations. This trend began with a slow effect, followed by several fluctuating stages in the middle, and finally stabilizing in the third stage. This demonstrates that end-of-pipe treatment of the air pollutant LAP can result in an increase in CO2 emissions. End-of-pipe treatment refers to the process of treating the pollutants generated by the consumption of energy, i.e. the use of desulfurization or dedusting technologies, e.g. in the production of materials for the construction of the LAP system (cement, steel), in the consumption of electricity for the use of the LAP system and in the chemical reactions during the LAP process, all of which can lead to an increase in CO2 emissions [32]. This indicates a desynchronization between pollution and carbon reduction. The CO2 emission of the YRD urban agglomeration shows a trend of first decreasing, then increasing, and then decreasing with the improvement of the industrial sulfur dioxide removal rate. The reasons are as follows: Improving desulphurization efficiency can effectively reduce the emission of harmful gases such as sulfur dioxide, thereby improving the atmospheric environment and slowing down the formation of acid rain. Secondly, adverse effects are caused by the final processing of LAP, as mentioned previously. Additionally, society as a whole will take appropriate countermeasures, such as optimizing energy use, improving production processes, and introducing more environmentally friendly technologies, to address the issue of increasing CO2 emissions. Introduction and implementation of a stricter environmental protection system, including the control of carbon emissions from industry and the energy sector, promotion of the development of renewable energy sources, encouragement of green scientific and technological innovation, and financial incentives for the low-carbon economy. These measures aim to further reduce CO2 emissions.

Industrial LAP emissions: The industrial sulfur dioxide removal rate and industrial fume and dust removal rate of LAP emissions in PRD urban agglomeration showed a three-stage multi-step downward trend. There were multiple fluctuation steps in the middle stage of development, followed by the third stage of regional stability.  This indicates that the reduction of LAP emissions is affected by the industrial sulfur dioxide removal rate and industrial fume and dust removal rate. The removal rate of industrial sulfur dioxide from the LAP emissions of the YRD urban agglomeration exhibited an 'inverted U' trend, suggesting that the reduction of LAP emissions was not affected by the industrial sulfur dioxide removal rate.  Similarly, the LAP emissions in the YRD urban agglomeration showed a three-stage, multi-step downward trend, with several fluctuation steps in the intermediate stage of development, before reaching the third stage of regional stability. The rate of industrial fume and dust removal affects the reduction of LAP emissions.

3.3.5. Trade effect factors-Total trade imports and exports (TR1)

Industrial CO2 emissions: Total trade imports and exports show a three-stage upward trend in CO2 emissions in both the PRD and the YRD agglomeration, from a slow initial effect to an intermediate stage that shows a steep rise and then to a steady state in the third stage. Total trade imports and exports, i.e. the sum of total trade imports and exports, can be used to observe the overall size of a country in terms of foreign trade. The above shows that as the total scale of foreign trade increases, CO2 emissions increase for two reasons: first, the increase in the total scale of foreign trade itself will promote the production activities of enterprises, which will lead to an increase in energy consumption and hence CO2 emissions; second, some foreign enterprises will transfer pollution-intensive industries to the PRD and YRD urban agglomeration through pollution transfer, which will lead to an increase in energy consumption and hence CO2 emissions [39].

Industrial LAP emissions: Total trade imports and exports show a three-stage downward trend in LAP emissions in both the PRD and YRD agglomerations, from a slow initial effect to a steep decline in the middle stage and a steady state in the third stage. The reason for the decrease in LAP emissions is that with the increase in the overall scale of foreign trade and the introduction of capital and technology from foreign-funded enterprises, the industrial sector can improve its production equipment to meet the requirements of "low energy consumption" and "high efficiency". "Some foreign-funded enterprises also choose to trade with enterprises that comply with local environmental technologies or regulations, thus encouraging more enterprises to improve their pollution reduction [27].

Point 6: In Table 4, the number of digits after the decimal point should be consistent.

Response 6: Thanks a lot for your comments. Your suggestions have been very helpful. We have corrected the number of decimal places in Table 4 to make sure they are consistent. This change is intended to improve the accuracy and readability of the table. The corrections are as follows:

original manuscript:

Lines 244-245:

Table 1. Importance of factors influencing industrial CO2 and LAP emissions

City Cluster

Scale effect factors

Structural effect factors

Technology effect factors

Governance Effect Factors

Trade effect factors

PRD

Indicators

EC1

ST1

TE1

TE2

EP1

EP2

EP3

SP1

LAP

0.1763

0.0455

0.2158

0.0499

0.0549

0.1594

0.1474

0.1506

CO2

0.1904

0.0323

0.2009

0.0486

0.0413

0.1783

0.1626

0.1456

YRD

Indicators

EC1

ST1

TE1

TE2

EP1

EP2

EP3

SP1

LAP

0.1676

0.1038

0.1171

0.1751

0.0319

0.1072

0.1800

0.1172

CO2

0.2077

0.0820

0.1964

0.05662

0.0199

0.1085

0.1591

0.1697

revised manuscript

Lines 289-291:

Table 2 Importance of factors influencing industrial CO2 and LAP emissions

Typology

CO2 and LAP emissions common main effect factor

CO2 emissions main effect factor

LAP emissions main effect factor

Importance of factors influencing

CO2&LAP0.125

CO20.125

LAP0.125

Color

The specific results of this study are shown in Table 5.

Table 3. Results on the importance of factors influencing industrial CO2 and LAP emissions

Region

Scale effect factors

Structural effect factors

Technology effect factors

Governance effect factors

Trade effect factors

PRD

Indicators

SC1

ST1

TE1

TE2

GO1

GO2

GO3

TR1

LAP

0.1652

0.1216

0.1681

0.0256

0.0227

0.1891

0.1427

0.1650

CO2

0.1780

0.0464

0.1915

0.1104

0.0328

0.1578

0.1498

0.1333

YRD

Indicators

SC1

ST1

TE1

TE2

GO1

GO2

GO3

TR1

LAP

0.1724

0.1481

0.1183

0.1862

0.0153

0.0906

0.1712

0.0979

CO2

0.1398

0.0533

0.1507

0.1415

0.1245

0.1169

0.1151

0.1581

Point 7: There is too much content in Section 4.2. Policy Implications and should be simplified.

Response 7: Thank you for the valuable suggestions. According to your suggestions, we have reformulated the contents of 4.2 Policy Implications, which are mainly divided into two major modules, namely, industrial sector and government sector, respectively, with implications on local conditions for the Pearl River Delta and Yangtze River Delta urban agglomerations. The goal of this revision is to better organize and present countermeasures and suggestions to make the article more structured and readable. We have put special emphasis on promoting the completion of the work on pollution reduction and carbon reduction in various aspects, so as to cover the relevant content more comprehensively. The simplifications are as follows:

original manuscript:

lines 481-522:

4.2. Policy Implications

(1) By analyzing the temporal dynamics of industrial CO2 and LAP emissions and intensity, we can see that both the PRD and the YRD urban agglomerations have better control of industrial LAP, but both need to further strengthen the control of industrial CO2, which means that the PRD and YRD urban agglomerations have greater potential for "carbon reduction". Therefore, both the PRD and the YRD urban agglomerations should focus more on "carbon reduction" and give full play to their synergistic effects in industrial pollution reduction and carbon reduction to help achieve the "double carbon" goal.

(2) Based on the analysis of the importance of factors influencing industrial CO2 and LAP emissions, both the PRD and the YRD urban agglomerations should pay attention to the influence of the scale effect. First, the industrial sector should conduct the analysis of economies of scale scientifically and efficiently; second, in the past, many industrial sectors adopted a sloppy economic growth mode, which also led to high resource consumption, slow capital turnover, losses and waste, low economic efficiency and other constant exist, and efforts should be made toward achieving an intensive economic growth mode. The PRD and YRD urban agglomerations should also pay attention to the impact of dust removal, "reduce pollution" and at the same time do not forget to "reduce carbon", ready to "catch both hands", not at the expense of industrial CO2 emissions At the cost of industrial LAP emissions, attention should be paid to the innovation of dust removal equipment and technology. Relevant departments should also improve the relevant environmental regulations and strictly enforce them.

(3) For the PRD urban agglomeration, attention should be paid to the planning of R&D funding and the pursuit of long-term effects, the same as dust removal treatment, attention should be paid to the innovation of desulfurization equipment and technology, continue to strengthen the role of the PRD urban agglomeration as a manufacturing center, continue to promote science and technology innovation strategy, further upgrade production equipment and technology, vigorously develop modern production services, and appropriately transfer pollution-intensive industries.

(4) For the Yangtze River Delta city cluster, attention should be paid to upgrading and transforming the industrial structure, optimizing the allocation of resources among industries, vigorously promoting science and technology innovation strategies, reducing highly polluting industries, increasing high-tech industries, and reducing energy consumption. The Yangtze River Delta city cluster, with Shanghai as the center, should vigorously develop tertiary industries such as finance, information and foreign trade and high-tech industries .The city group of Yangtze River Delta, with Shanghai as the center, should vigorously develop tertiary industries such as finance, information and foreign trade and high-tech industries, and can give certain preferential treatment to low-pollution and high-tech foreign-funded enterprises to promote economic growth while introducing technology. The Yangtze River Delta city cluster should also pay attention to the planning of environmental protection investment, strengthen the promotion of environmental protection concepts, support the innovation of green technology, and invest more proportion of the funds in the construction of ecological civilization.

revised manuscript

Lines 551-578:

4.2. Policy Implications

(1) By analyzing the time-dynamic evolution of industrial CO2 and LAP emissions and intensities, it can be seen that the PRD and YRD urban agglomeration are both more effective in managing industrial LAP. However, the management of industrial CO2 still requires further strengthening. Therefore, both the PRD and YRD urban agglomerations should focus more on the 'Carbon Reduction' perspective in their efforts to reduce carbon emissions. Therefore, the PRD and YRD should prioritize 'carbon reduction' and maximize their synergistic effects in reducing industrial CO2 emissions to help achieve the 'dual carbon' goal.

(2) Based on the analysis of the importance of factors influencing industrial CO2 and LAP emissions, both PRD urban agglomeration and YRD urban agglomeration should pay attention to the influence of the scale effect. The industrial sector should first scientifically and efficiently analyze the benefits of economies of scale. Secondly, in the past, many industrial sectors have adopted a careless approach to economic growth, resulting in excessive resource consumption, slow capital turnover, high losses and wastage, low economic efficiency, and so on. We should strive to achieve intensive economic growth.

In the PRD urban agglomeration, the industrial sector should focus on dust and sulfur removal, as well as promoting equipment and technological innovation. Government sector can promote the impact of foreign trade on pollution and carbon reduction by formulating environmental protection standards, engaging in international cooperation, incentivizing the adoption of cleaner technologies, and strengthening regulation to achieve a balance between sustainable economic development and environmental protection.

For the YRD urban agglomeration, the industrial sector should pay attention to the upgrading and transformation of the industrial structure, optimize the allocation of resources among industries, vigorously promote the strategy of scientific and technological innovation, reduce the number of highly polluting industries, increase the number of high-tech industries and reduce energy consumption.

Point 8: There are some syntax errors, sentence errors and other English Language errors that must be corrected, e.g. “for example, Tang Xiangbo et al.[6] For example, Tang Xiangbo et al.”; Line 111, “In this thesis”. Possibly, a native English Language speaking Scientists should be employed for the final editing.

Response 8: Thanks a lot for your comments and recommendations. We are very sorry for some grammatical and expression errors. After careful verification, we have corrected them one by one, and will check the article more carefully to enhance the rigor of the article. The specific corrections are as follows:

original manuscript:

lines 61-82 :

By reviewing the relevant literature, scholars have conducted in-depth studies on the influencing factors of pollution reduction and carbon reduction from different perspectives in terms of achieving the goal of pollution reduction and carbon reduction. The research level perspectives can be summarized at the national level[1] local level[2], industry sector level[3], etc. The methodological perspective can be summarized as the decomposition analysis method, which includes two methods: index decomposition analysis (IDA) and structural decomposition analysis (SDA). For example, Huang Ru Xia et al.[4] (2004) used the structural decomposition analysis method to explore the mechanism of the effect of industrial structure change on synergistic pollution reduction and carbon reduction. Jiang, Yuqing, et al.[5] used the structural decomposition method to analyze the drivers of pollution and carbon reduction in China. In addition, some scholars have also used econometric models to study the influencing factors of pollution reduction and carbon reduction, for example, Tang Xiangbo et al.[6]  For example, Tang Xiangbo et al. investigated the spatial evolution pattern and mechanism of the synergistic effect of pollution reduction and carbon reduction based on provincial data in China through a spatiotemporal geographically weighted regression model, and Liu Maohui et al.[7] analyzed the synergistic effect of pollution reduction and carbon reduction in Tianjin based on a spatial error model. Li Zijie et al.[8] studied the main factors influencing carbon emissions in the Yangtze River Economic Zone through a spatiotemporal geographically weighted regression model. In the index selection, scholars mainly attributed the influencing factors of pollution reduction and carbon reduction to the following aspects: scale effect, structural effect, technology effect, governance factor, trade factor and population factor.[9]

revised manuscript:

Lines 67-85:

The research has been conducted at the national level [4-7], the local level [8,9], and the industry sector level [10,11]. The research method perspective is mainly attributed to the decomposition analysis method, which includes the index decomposition analysis method (IDA) and the structural decomposition analysis method (SDA) [12,13]. For instance, Yan et al. (2019) used the IDA method to identify the main influencing factors of the overall change in CO2 emissions from provincial thermal power generation in China [14]. Wang et al. (2019) employed the SDA method to investigate the driving factors behind changes in carbon emissions resulting from domestic trade in China at a regional scale [15]. Other scholars have also utilized econometric models to examine the factors that influence pollution and carbon reduction [16]. For instance, Xian et al. (2019) evaluated the synergistic effect and mechanism of the carbon trading pilot policy on carbon and air pollutant emissions in China using the double difference model (DID). The study covered the power, industry, transportation, and residential sectors [17]. Zhu et al. (2020) analyzed the factors influencing hidden carbon emissions in China's construction industry using the extended STIRPAT mode [18]. Lin et al. (year not provided) used the Spatial Durbin Error Model (SDEM) to investigate the socioeconomic factors that can jointly impact CO2 and PM2.5 emissions [19]. Scholars have identified the following aspects as the main influencing factors of pollution and carbon reduction: scale effect, structural effect, technical effect, governance factors, trade factors, and population factors [20-22].

original manuscript:

lines 111-113:

In this thesis, a total of five fossil energy sources, namely gasoline, diesel Oil, raw coal, coke and natural gas, were selected to measure the CO2 emissions from the combustion of the above fossil energy sources using the emission factor method.

revised manuscript

Lines 136-138:

This paper examines five fossil energy sources: gasoline, diesel oil, raw coal, coke, and natural gas. The CO2 emissions resulting from the combustion of these sources are measured using the emission factor method.

Point 9: The format of references should be consistent, e.g. Line 443, 464, 490, 516, 522, 533, 544, the page number is missed. Line 548, “Huang Ruxia, Zhong Qiumeng, Wu Xiaohui”; Line 558, “Li Z.J., Xu J.L., Wang J”; Line 596, “FRONTIERS OF ENVIRONMENTAL SCIENCE & ENGINEERING”; etc.

Response 9: Thank you very much for your careful review and feedback on the format of the references. We have revised the format of all documents in accordance with the required format of the journal, ensuring the consistency of the format of the references, and filling in relevant information such as missing page numbers and DOI. This revision aims to improve the overall normativity and credibility of the paper. The specific corrections are as follows:

Lines 594-688:

References

  1. Yuan, X.; Su, C.-W.; Umar, M.; Shao, X.; LobonŢ, O.-R. The race to zero emissions: Can renewable energy be the path to carbon neutrality? Journal of Environmental Management 2022, 308, 114648, doi:10.1016/j.jenvman.2022.114648.
  2. Vara Prasad, M.N.; Smol, M.; Freitas, H. Chapter 1 - Achieving sustainable development goals via green deal strategies. In Sustainable and Circular Management of Resources and Waste Towards a Green Deal, Vara Prasad, M.N., Smol, M., Eds.; Elsevier: 2023; pp. 3-23.doi: 10.1016/B978-0-323-95278-1.00002-4
  3. Chaturvedi, V.; Koti, P.N.; Chordia, A.R. Pathways towards India's nationally determined contribution and mid-century strategy. Energy and Climate Change 2021, 2, 100031, doi:10.1016/j.egycc.2021.100031.
  4. Chen, X.H.; Tee, K.; Elnahass, M.; Ahmed, R. Assessing the environmental impacts of renewable energy sources: A case study on air pollution and carbon emissions in China. Journal of Environmental Management 2023, 345, 118525, doi:10.1016/j.jenvman.2023.118525.
  5. Zhang, Z.H.; Zhang, J.X.; Feng, Y.C. Assessment of the Carbon Emission Reduction Effect of the Air Pollution Prevention and Control Action Plan in China. International Journal of Environmental Research and Public Health 2021, 18, 13307, doi:10.3390/ijerph182413307.
  6. Akter, S.; Bennett, J. Household perceptions of climate change and preferences for mitigation action: the case of the Carbon Pollution Reduction Scheme in Australia. Climatic Change 2011, 109, 417-436, doi:10.1007/s10584-011-0034-8.
  7. Wang, S.J.; Zhang, S.H.; Cheng, L. Investigating the Synergy between CO2 and PM2.5 Emissions Reduction: A Case Study of China's 329 Cities. Atmosphere 2023, 14, doi:10.3390/atmos14091338.
  8. Chen, S.Y.; Tan, Z.X.; Mu, S.Y.; Wang, J.Y.; Chen, Y.Y.; He, X.W. Synergy level of pollution and carbon reduction in the Yangtze River Economic Belt: Spatial-temporal evolution characteristics and driving factors. Sustainable Cities and Society 2023, 98, 104859, doi:10.1016/j.scs.2023.104859.
  9. Hu, C.; Ma, X.; Yang, L.; Chang, X.; Li, Q. Spatial-temporal variation and driving forces of the synergy of “pollution reduction, carbon reduction, green expansion and economic growth”: evidence from 243 cities in China. 2023, 11, 1-28, doi:10.3389/fevo.2023.1202898.
  10. Zhao, J.Y.; Xi, X.; Na, Q.; Wang, S.S.; Kadry, S.N.; Kumar, P.M. The technological innovation of hybrid and plug-in electric vehicles for environment carbon pollution control. Environmental Impact Assessment Review 2021, 86, 106506, doi:10.1016/j.eiar.2020.106506.
  11. Jiang, Y.; Zhang, Y.; Brenya, R.; Wang, K. How environmental decentralization affects the synergy of pollution and carbon reduction: Evidence based on pig breeding in China. Heliyon 2023, 9, e21993, doi:10.1016/j.heliyon.2023.e21993.
  12. Andreoni, V.; Galmarini, S. Drivers in CO2 emissions variation: A decomposition analysis for 33 world countries. Energy 2016, 103, 27-37, doi:10.1016/j.energy.2016.02.096.
  13. Wang, S.J.; Zhang, S.H.; Cheng, L. Drivers and Decoupling Effects of PM2.5 Emissions in China: An Application of the Generalized Divisia Index. International Journal of Environmental Research and Public Health 2023, 20, doi:10.3390/ijerph20020921.
  14. Yan, Q.Y.; Wang, Y.X.; Balezentis, T.; Sun, Y.K.; Streimikiene, D. Energy-Related CO2 Emission in China's Provincial Thermal Electricity Generation: Driving Factors and Possibilities for Abatement. Energies 2018, 11, 1096, doi:10.3390/en11051096.
  15. Wang, Z.H.; Li, Y.M.; Cai, H.L.; Yang, Y.T.; Wang, B. Regional difference and drivers in China's carbon emissions embodied in internal trade. Energy Economics 2019, 83, 217-228, doi:10.1016/j.eneco.2019.06.023.
  16. Wang, Z.; Shao, H. Spatiotemporal differences in and influencing factors of urban carbon emission efficiency in China’s Yangtze River Economic Belt. Environmental Science Pollution Research 2023, 30, 121713-121733, doi:10.1007/s11356-023-30674-7.
  17. Xian, B.; Wang, Y.; Xu, Y.; Wang, J.; Li, X. Assessment of the co-benefits of China's carbon trading policy on carbon emissions reduction and air pollution control in multiple sectors. Economic Analysis and Policy 2024, doi:10.1016/j.eap.2024.01.011.
  18. Zhu, C.; Chang, Y.; Li, X.D.; Shan, M.Y. Factors influencing embodied carbon emissions of China's building sector: An analysis based on extended STIRPAT modeling. Energy and Buildings 2022, 255, 14, doi:10.1016/j.enbuild.2021.111607.
  19. Lin, H.X.; Jiang, P. Analyzing the phased changes of socioeconomic drivers to carbon dioxide and particulate matter emissions in the Yangtze River Delta. Ecological Indicators 2022, 140, 109044, doi:10.1016/j.ecolind.2022.109044.
  20. Zhang, P.D. End-of-pipe or process-integrated: evidence from LMDI decomposition of China's SO2 emission density reduction. Frontiers of Environmental Science & Engineering 2013, 7, 867-874, doi:10.1007/s11783-013-0541-0.
  21. Zha, Q.F.; Liu, Z.; Wang, J. Spatial pattern and driving factors of synergistic governance efficiency in pollution reduction and carbon reduction in Chinese cities. Ecological Indicators 2023, 156, 111198, doi:10.1016/j.ecolind.2023.111198.
  22. Du, Q.; Pang, Q.Y.; Bao, T.N.; Guo, X.Q.; Deng, Y.G. Critical factors influencing carbon emissions of prefabricated building supply chains in China. Journal of Cleaner Production 2021, 280, 12, doi:10.1016/j.jclepro.2020.124398.
  23. Liu, M.Z.; Zhang, X.X.; Zhang, M.Y.; Feng, Y.Q.; Liu, Y.J.; Wen, J.X.; Liu, L.Y. Influencing factors of carbon emissions in transportation industry based on C-D function and LMDI decomposition model: China as an example. Environmental Impact Assessment Review 2021, 90, 106623, doi:10.1016/j.eiar.2021.106623.
  24. MAO X, X.Y., GAO Y,HE F, ZENG A,KUAI P,HU T. Study on GHGs and air pollutants co-control: assessment and planning. 2021, 41, 3390-3398, doi:10.19674/j.cnki.issn1000-6923.2021.0316.
  25. Cheng, Z.H.; Li, L.S.; Liu, J. Industrial structure, technical progress and carbon intensity in China's provinces. Renewable & Sustainable Energy Reviews 2018, 81, 2935-2946, doi:10.1016/j.rser.2017.06.103.
  26. Chen, X.; Di, Q.; Jia, W.; Hou, Z. Spatial correlation network of pollution and carbon emission reductions coupled with high-quality economic development in three Chinese urban agglomerations. Sustainable Cities and Society 2023, 94, 104552, doi:10.1016/j.scs.2023.104552.
  27. Zeng, S.L.; Liu, Y.Q.; Ding, J.J.; Xu, D.L. An Empirical Analysis of Energy Consumption, FDI and High Quality Development Based on Time Series Data of Zhejiang Province. International Journal of Environmental Research and Public Health 2020, 17, 3321, doi:10.3390/ijerph17093321.
  28. Liu, Z.; Meng, J.; Deng, Z.; Lu, P.; Guan, D.B.; Zhang, Q.; He, K.B.; Gong, P. Embodied carbon emissions in China-US trade. Science China-Earth Sciences 2020, 63, 1577-1586, doi:10.1007/s11430-019-9635-x.
  29. Fujii, H.; Managi, S.; Kaneko, S. Decomposition analysis of air pollution abatement in China: empirical study for ten industrial sectors from 1998 to 2009. Journal of Cleaner Production 2013, 59, 22-31, doi:10.1016/j.jclepro.2013.06.059.
  30. Lyu, W.; Li, Y.; Guan, D.B.; Zhao, H.Y.; Zhang, Q.; Liu, Z. Driving forces of Chinese primary air pollution emissions: an index decomposition analysis. Journal of Cleaner Production 2016, 133, 136-144, doi:10.1016/j.jclepro.2016.04.093.
  31. Zhou, D.; Zhang, X.R.; Wang, X.Q. Research on coupling degree and coupling path between China's carbon emission efficiency and industrial structure upgrading. Environmental Science and Pollution Research 2020, 27, 25149-25162, doi:10.1007/s11356-020-08993-w.
  32. Li M, W.Q. Study on the Technology Progress Impact on Pollutant Generation Based on Malmquist Index: Take Industry SO2 as an Example. Acta Scientiarum Naturalium Unversitatis Pekinensis 2012, 48, 817-823, doi:10.13209/j.0479-8023.2012.106.
  33. Liu Q, W.Q., Liu Y. . Study on the relationship between economic growth, international trade and pollution emissions: an empirical analysis based on SO2 emissions in the United States and China. China Population, Resources and Environment 2012, 22, 170-176, doi:10.3969/j.issn.1002-2104.2012.05.028.
  34. Breiman, L. Random Forests. Machine Learning 2001, 45, 5-32, doi:10.1023/A:1010933404324.
  35. Kulkarni, S.S.; Sanghai, N.P.; Baishya, C.; Kumar, A.; Nayak, S.K. Random forest regression for radiation pattern prediction of planar metasurface reflector antenna. Aeu-International Journal of Electronics and Communications 2024, 174, 155018, doi:10.1016/j.aeue.2023.155018.
  36. Yu W, Z.T., Shen D. . County-levelspatial pattern and influencing factors evolution of carbon emission intensity in China: A random forest model analysis. . China Environmental Science 2022, 42, 2788-2798, doi:10.19674/j.cnki.issn1000-6923.20220219.001.
  37. Xu K, W.Y. The Influence of China’s OFDI on Its Domestic CO2 Emissions: An Empirical Analysis Based on China’s Provincial Panel Data. GUOJI SHANGWU YANJIU 2015, 76-86, doi:10.13680/j.cnki.ibr.2015.01.008.
  38. Shi, J.; Huang, W.; Han, H.; Xu, C. Pollution control of wastewater from the coal chemical industry in China: Environmental management policy and technical standards. Renewable and Sustainable Energy Reviews 2021, 143, 110883, doi:10.1016/j.rser.2021.110883.
  39. Copeland, B.R.; Taylor, M.S. Trade, growth, and the environment. Journal of Economic literature 2004, 42, 7-71, doi:10.1257/002205104773558047.

Point 10: Moderate editing of English language required.

Response 10: Thank you very much for your kind and constructive comments on our manuscript. They have helped to significantly improve the quality of the manuscript. According to your suggestion, we have corrected the grammatical and editorial mistakes throughout the manuscript. Moreover, we have invited an English native speaker to improve the English language of the revised manuscript. All the changes have been highlighted in yellow in the revised manuscript.

original manuscript:

lines 34-98:

Energy and environmental issues have become important factors in global economic and social development, and taking the initiative to address climate change and develop a low-carbon economy has become an international consensus. As the largest energy consumer and carbon emitter, China has a heavy task to reduce pollution and carbon emissions. On September 22, 2020, Xi Jinping, General Secretary of the CPC Central Committee, President of the People's Republic of China and Chairman of the Central Military Commission, delivered an important speech at the general debate of the 75th session of the UN General Assembly, announcing that China's carbon dioxide emissions will strive to peak by 2030 and strive to achieve carbon neutrality by 2060. This is the first time that China has made a commitment on the UN stage, and this commitment is of great significance to global climate governance. On February 26, 2021, the National Development and Reform Commission (NDRC) and the State-owned Assets Supervision and Administration Commission of the State Council (SASAC) issued the "Guidance on Strengthening Green and Low-carbon Transformation and Development", which proposed to achieve a peak in CO2 emissions by 2030. The "Double Carbon" target aims to achieve a peak in carbon dioxide emissions by 2030 and will strive to reach the peak as soon as possible and to reduce carbon dioxide emissions per unit of GDP by more than 65% compared to 2005. The "double carbon" target aims to promote China's high-quality development and implement ecological civilization construction. The Pearl River Delta (PRD) and Yangtze River Delta (YRD) urban agglomerations are two economically developed and highly industrialized regions and are also one of the major manufacturing bases in China. How to achieve the "double carbon" goal by reducing pollution and carbon has become a major issue for these two regions. In this paper, we study the factors influencing industrial pollution and carbon reduction in the PRD and YRD urban agglomerations, and propose a mechanism for coordinated regional development under the "double carbon" target, in order to achieve collaborative industrial development, technological innovation and green low-carbon transformation.

By reviewing the relevant literature, scholars have conducted in-depth studies on the influencing factors of pollution reduction and carbon reduction from different perspectives in terms of achieving the goal of pollution reduction and carbon reduction. The research level perspectives can be summarized at the national level [1] local level[2], industry sector level[3], etc. The methodological perspective can be summarized as the decomposition analysis method, which includes two methods: index decomposition analysis (IDA) and structural decomposition analysis (SDA). For example, Huang Ru Xia et al.[4] (2004) used the structural decomposition analysis method to explore the mechanism of the effect of industrial structure change on synergistic pollution reduction and carbon reduction. Jiang, Yuqing, et al.[5] used the structural decomposition method to analyze the drivers of pollution and carbon reduction in China. In addition, some scholars have also used econometric models to study the influencing factors of pollution reduction and carbon reduction, for example, Tang Xiangbo et al.[6]  For example, Tang Xiangbo et al. investigated the spatial evolution pattern and mechanism of the synergistic effect of pollution reduction and carbon reduction based on provincial data in China through a spatiotemporal geographically weighted regression model, and Liu Maohui et al.[7] analyzed the synergistic effect of pollution reduction and carbon reduction in Tianjin based on a spatial error model. Li Zijie et al.[8] studied the main factors influencing carbon emissions in the Yangtze River Economic Zone through a spatiotemporal geographically weighted regression model. In the index selection, scholars mainly attributed the influencing factors of pollution reduction and carbon reduction to the following aspects: scale effect, structural effect, technology effect, governance factor, trade factor and population factor.[9]

On the whole, although the existing research has laid the foundation for reducing pollution and carbon, and has achieved some results, there are still some shortcomings. First of all, the reduction of pollution and carbon at the industrial level is a key point of the "double carbon" work, but the existing studies still pay little attention to this aspect, and most of the results only focus on the influence factors of greenhouse gas or air pollutant reduction, which cannot meet the existing development. Moreover, up to now, the research on the influence factors of pollution reduction and carbon reduction in the PRD and YRD urban agglomerations has not been sufficiently explored, and the research on inter-provincial heterogeneity is insufficient to provide management decision support for local governments to formulate differentiated policies on pollution reduction and carbon reduction.

Based on this, this thesis adopts the random forest regression model method, selects the PRD and YRD urban agglomerations as the study area, and identifies the main influencing factors of industrial pollution reduction and carbon reduction by analyzing the data of industrial CO2 and industrial LAP from 2000 to 2020 and gives related countermeasure suggestions.

revised manuscript

Lines 32-123:

Energy and environmental issues are crucial factors in global economic and social development. There is an international consensus to address climate change and develop a low-carbon economy proactively. China, being the largest country in energy consumption and carbon emissions, has a significant responsibility to reduce pollution and carbon emissions. On 22 September 2020, Xi Jinping, General Secretary of the Communist Party of China Central Committee, Chinese President and Chairman of the Central Military Commission, delivered a speech at the general debate of the 75th United Nations General Assembly. He announced that China aims to peak carbon dioxide emissions by 2030 and achieve carbon neutrality by 2060. This marks China's inaugural commitment to the United Nations stage, a significant development in global climate governance. The 'dual-carbon' goal was subsequently incorporated into China's 14th Five-Year Plan and 2035 Vision Outline. On February 26th, 2021, the National Development and Reform Commission (NDRC), the State-owned Assets Supervision and Administration Commission of the State Council (SASAC), and others issued the 'Guiding Opinions on Strengthening Green and Low-Carbon Transformation and Development.' The goal proposed is to achieve peak carbon dioxide emissions before 2030. The 'dual-carbon' goal aims to promote China's high-quality development and the construction of ecological civilization. This concept emphasizes the harmonious coexistence of human beings with nature, sustainable development, and environmental protection. It is a comprehensive development concept in China. Additionally, countries around the world are implementing various 'dual-carbon' policies to address climate change and achieve carbon peak and neutrality. For instance, in the United States, these policies involve rejoining the Paris Agreement, setting ambitious carbon-neutral targets for 2050, and making significant investments in clean energy and infrastructure. The administration's policy emphasizes comprehensive climate action, including concerns about specific sector objectives, policy reversals, and cooperation with domestic and international partners [1]. The European Union's 'Green Deal' is an ambitious plan to transform its member states into carbon-neutral economies by 2050. This plan addresses the energy transition, promotion of renewable energy, and industrial innovation [2]. Despite its energy-intensive industries, India is aggressively pursuing carbon reduction targets and investing in renewable energy and clean technologies for sustainable development [3]. These policies reflect the global consensus on cooperation in tackling climate change and highlight the challenges faced by countries in achieving carbon neutrality.

Scholars have researched the factors influencing pollution and carbon reduction from various perspectives to achieve the goal of reducing pollution and carbon emissions. The research has been conducted at the national level [4-7], the local level [8,9], and the industry sector level [10,11]. The research method perspective is mainly attributed to the decomposition analysis method, which includes the index decomposition analysis method (IDA) and the structural decomposition analysis method (SDA) [12,13]. For instance, Yan et al. (2019) used the IDA method to identify the main influencing factors of the overall change in CO2 emissions from provincial thermal power generation in China [14]. Wang et al. (2019) employed the SDA method to investigate the driving factors behind changes in carbon emissions resulting from domestic trade in China at a regional scale [15]. Other scholars have also utilized econometric models to examine the factors that influence pollution and carbon reduction [16]. For instance, Xian et al. (2019) evaluated the synergistic effect and mechanism of the carbon trading pilot policy on carbon and air pollutant emissions in China using the double difference model (DID). The study covered the power, industry, transportation, and residential sectors [17]. Zhu et al. (2020) analyzed the factors influencing hidden carbon emissions in China's construction industry using the extended STIRPAT mode [18]. Lin et al. (year not provided) used the Spatial Durbin Error Model (SDEM) to investigate the socioeconomic factors that can jointly impact CO2 and PM2.5 emissions [19]. Scholars have identified the following aspects as the main influencing factors of pollution and carbon reduction: scale effect, structural effect, technical effect, governance factors, trade factors, and population factors [20-22].

In general, the following shortcomings persist: (1) The reduction of pollution and carbon emissions at the industrial level is a crucial aspect of the 'dual-carbon' program. However, existing research has largely neglected this area, with most studies focusing on the factors that influence the reduction of greenhouse gases or air pollutants. This is insufficient to meet current development needs. (2) To date, research on the factors that influence the reduction of pollution and carbon in the urban agglomerations of the Pearl River Delta and Yangtze River Delta has been inadequate. Further exploration of the YRD urban agglomeration is necessary, along with additional research on inter-provincial heterogeneity. This will provide local governments with the necessary information to formulate differentiated pollution and carbon reduction policies.

Therefore, we analyzed data on industrial CO2 and LAP from 2002 to 2021 and used a random forest regression model to systematically study the factors influencing pollution and carbon reduction in the PRD and YRD urban agglomerations. The main objectives are: (1) The aim of this study is to identify the main factors that influence industrial pollution reduction and carbon reduction in the PRD and YRD urban agglomerations. (2) Additionally, we aim to explore the temporal dynamic evolution patterns of industrial CO2 and LAP emissions and intensities in these areas. (3) The results of this study will provide a more comprehensive and precise reference for the formulation of environmental protection policies for the PRD and YRD urban agglomerations. This approach contributes to a comprehensive understanding of the interactions between various factors in pollution and carbon reduction. It provides a scientific basis for future sustainable urban development.

This paper presents three new contributions. Firstly, it discusses pollution reduction and carbon reduction in the same framework. Secondly, it highlights the significance of fully utilizing the synergistic effects of the PRD and YRD urban agglomerations in industrial pollution reduction and carbon reduction. Finally, it emphasizes that the PRD and YRD urban agglomerations play a key role in China's economic and social development while simultaneously facing unique environmental challenges. This choice takes into account several factors, including the concentration of economic activities, differences in energy mixes, geographical characteristics, and potential policy implementation. The two regions share similarities in terms of high economic development and urbanization, but differ in their geographic location and industrial composition. Additionally, this thesis aims to provide information on the similarities and differences between the selected clusters in terms of industrial CO2 and LAP emission reductions, while acknowledging the potential inter-provincial heterogeneity of the two clusters, including differences in environmental policies, resource distribution, and economic status. Thirdly, it provides a methodology to investigate how to achieve synergistic development of pollution reduction and carbon reduction in multiple aspects, including energy, industrial structure, production technology, science, technology and innovation, and trade.
